# PROJECTION HEAD IS SECRETLY AN INFORMATION BOTTLENECK

**Zhuo Ouyang**[1*]     **Kaiwen Hu**[2*]     **Qi Zhang**[3]     **Yifei Wang**[4]     **Yisen Wang**[3,5†]

[1] College of Engineering, Peking University
[2] School of EECS, Peking University
[3] State Key Lab of General Artificial Intelligence,
   School of Intelligence Science and Technology, Peking University
[4] MIT CSAIL
[5] Institute for Artificial Intelligence, Peking University

## ABSTRACT

Recently, contrastive learning has risen to be a promising paradigm for extracting meaningful data representations. Among various special designs, adding a projection head on top of the encoder during training and removing it for downstream tasks has proven to significantly enhance the performance of contrastive learning. However, despite its empirical success, the underlying mechanism of the projection head remains under-explored. In this paper, we develop an in-depth theoretical understanding of the projection head from the information-theoretic perspective. By establishing the theoretical guarantees on the downstream performance of the features before the projector, we reveal that an effective projector should act as an information bottleneck, filtering out the information irrelevant to the contrastive objective. Based on theoretical insights, we introduce modifications to projectors with training and structural regularizations. Empirically, our methods exhibit consistent improvement in the downstream performance across various real-world datasets, including CIFAR-10, CIFAR-100, and ImageNet-100. We believe our theoretical understanding on the role of the projection head will inspire more principled and advanced designs in this field. Code is available at `https://github.com/PKU-ML/Projector_Theory`.

## 1 INTRODUCTION

In recent years, contrastive learning has emerged as a promising representation learning paradigm and exhibited impressive performance without supervised labels (Chen et al., 2020; He et al., 2020; Zbontar et al., 2021). The core idea of contrastive learning is quite simple, that is to pull the augmented views of the same samples (i.e., positive samples) together while pushing the independent samples (i.e., negative samples) away. To improve the downstream performance of contrastive learning, researchers have proposed various special training objectives and architecture designs (Grill et al., 2020; Wang et al., 2021; Guo et al., 2023; Wang et al., 2023; 2024; Du et al., 2024). Among them, one of the most widely-used techniques is the projection head (i.e., projector) (Chen et al., 2020), which is a shallow layer following the backbone during pretraining and is discarded in downstream tasks like image classification and object detection. It has been shown that the features before the projector (denoted as encoder features) exhibit much better downstream performance than the features after the projector (denoted as projector features) across various applications (Jing et al., 2021; Gupta et al., 2022). Inspired by the success of the projection head in contrastive learning, researchers also extend this architecture to other representation learning paradigms and achieve significant improvements (Sariyildiz et al., 2022; Zhou et al., 2021). However, although the projection head has been widely adopted, the understanding of the underlying mechanism behind it is still quite limited. In this paper, we aim to establish a theoretical analysis of the relationship between the projection head and the downstream performance of contrastive learning.

Among various theoretical understandings of contrastive learning (Arora et al., 2019; Oord et al., 2018; HaoChen et al., 2021; Wang et al., 2022; Cui et al., 2023), information theory provides a

---

[*]Equal Contribution.
[†]Corresponding Author: Yisen Wang (yisen.wang@pku.edu.cn).

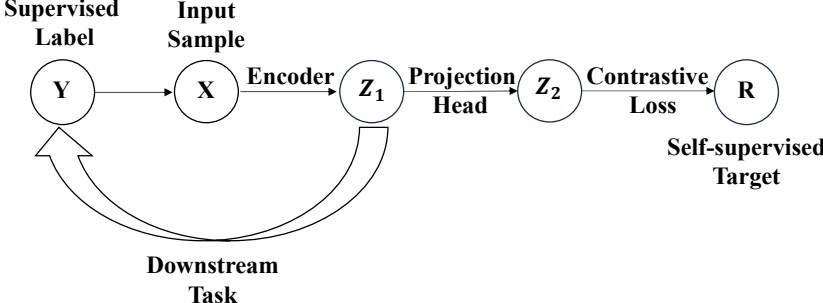

Figure 1: The general construction of contrastive learning can be displayed as the information flow model above, where $Y$ denote the ground-truth labels in downstream tasks, $X$ are the input samples, $Z_1, Z_2$ denote the encoder and projector features, and $R$ represent the self-supervised targets.

useful framework for analyzing its downstream performance (Oord et al., 2018; Tian et al., 2020). Specifically, as shown in Figure 1, the overall information flow in contrastive learning transmits from $Y$ to $R$, where $Y$ represents the natural semantics that generates input samples $X$ (e.g., the supervised labels), $Z_1$ (encoder features) and $Z_2$ (projector features) respectively denote the features before and after the projector, and $R$ represents the self-supervised targets in contrastive learning. It has been proven that neural networks can capture most of the essential information needed for downstream tasks by minimizing the contrastive objective, i.e., maximizing the mutual information $I(Z_2; R)$ leads to maximizing $I(Z_2; Y)$ (Tian et al., 2020). However, a crucial question remains: *what is the influence of discarding the projection head in downstream tasks?*

By establishing theoretical guarantees, we identify some critical factors affecting the downstream performance of the encoder features. Specifically, the downstream utility of encoder features, quantified by $I(Z_1; Y)$, improves as the mutual information between encoder features and projector features $I(Z_1; Z_2)$ decreases while the mutual information between encoder features and the contrastive objective $I(Z_1; R)$ increases. To balance these two factors, our theoretical analysis suggests an essential design principle for the projector: it should function as an information bottleneck, selectively filtering out information that is irrelevant to the contrastive objective.

Based on the theoretical principle, we propose two approaches to optimize the designs of projectors, i.e., modifying the training objective and adjusting the projector architectures. To be specific, for controlling the mutual information between encoder and projector features ($I(Z_1; Z_2)$) with new training objectives, we estimate the mutual information with the surrogate metric proposed by Tan et al. (2023) and incorporate it as a regularization term. For structural adjustments, we decrease the mutual information between encoder and projector features ($I(Z_1; Z_2)$) by promoting the sparsity of the projector outputs with discretization (Liu et al., 2002) and sparse autoencoder (Ng et al., 2011) architectures. With the proposed improvements on the projectors, our methods exhibit significantly improved downstream performance across various real-world datasets and contrastive frameworks. Our contributions are summarized as follows:

- We develop a new theoretical understanding for the role of the projection head in contrastive learning from the information-theoretic perspective. We establish both lower and upper bounds for the downstream performance of features before the projector.
- Our findings indicate theoretical principles for designing an effective projection head: it should act as an information bottleneck, filtering out the irrelevant information and preserving the essential information for the contrastive objective.
- Based on theoretical principles, we propose two categories of methods to improve projector design, namely training regularization and structural regularization. Various experiments on different datasets and contrastive methods demonstrate that our modified projectors significantly improve the downstream performance of contrastive learning.

## 2 RELATED WORK

**Contrastive Learning.** Recent advancements in contrastive learning algorithms have revolutionized the field of representation learning by leveraging the principle of instance discrimination. Represen-

tative algorithms such as SimCLR (Chen et al., 2020) and MoCo (He et al., 2020) have demonstrated the power of contrastive approaches by maximizing alignment between differently augmented views of the same data and minimizing similarity between different instances. The following frameworks including BYOL (Grill et al., 2020), Barlow Twins (Zbontar et al., 2021), SimSiam (Chen & He, 2021) extend contrastive learning with different objectives, structures, and empirical tricks. Among various designs in contrastive learning, an essential component is the projection head, which employs a multi-layer perceptron (MLP) on top of the backbone network during the pretraining process and discards it in the downstream task. The projection head has been found to be crucial in improving the quality of learned representations. The features before the projection head exhibit benefits in generalization (Chen et al., 2020), robustness (Gupta et al., 2022), and avoiding dimensional collapse (Jing et al., 2021). Due to the impressive performance of the projection head, researchers extend this structure to different self-supervised paradigms (Zhou et al., 2021; Zhang et al., 2022; Yue et al., 2023; Zhang et al., 2023), and even to supervised learning (Sariyildiz et al., 2022), leading to significant improvements across various scenarios. However, although the projection head has achieved impressive empirical success, the theoretical understanding remains quite limited.

**Theory of Contrastive Learning.** As contrastive learning algorithms have achieved remarkable empirical success across various downstream applications, researchers try to understand the mechanisms behind them from different theoretical perspectives. Wang & Isola (2020) understand contrastive objectives by reformulating the InfoNCE loss as two terms: the alignment of positive pairs and the uniformity of negative samples. Arora et al. (2019) establish the first theoretical guarantee between the contrastive loss and the downstream classification loss. HaoChen et al. (2021) evaluate the downstream performance of the optimal solution in contrastive learning by connecting the contrastive objective to matrix factorization. Saunshi et al. (2022) point out that understanding contrastive learning methods requires incorporating inductive biases. Besides, several works discuss the selection of positive samples (Tian et al., 2020), introduce new training objectives (Bardes et al., 2021), and analyze the learning dynamics (Tan et al., 2023; Wang et al., 2023). Additionally, for analyzing the special architectures in contrastive learning, previous works establish theoretical analysis of the prediction head in contrastive learning without negative samples (Wen & Li, 2022; Zhuo et al., 2023), investigate the effect of the projection head in both supervised and self-supervised learning (Xue et al., 2024) and apply the concepts of expansion and shrinkage to explain the effectiveness of projection head (Gui et al., 2023).

## 3 AN INFORMATION-THEORETIC ANALYSIS ON THE ROLE OF PROJECTORS

Among various theoretical frameworks for contrastive learning (Arora et al., 2019; HaoChen et al., 2021; Oord et al., 2018; Wang et al., 2022), the information-theoretic perspective is popular and offers many insightful understandings (Tian et al., 2020; Tan et al., 2023). However, previous analyses usually overlook the existence of the projection head and discuss the downstream performance of projector features, which leads to an obvious gap between theory and practice. In this paper, we still use the information-theoretic perspective, but focus on providing a theoretical understanding of encoder features.

In this section, we start with an introduction to the information flow in contrastive learning to lay a foundation for our analysis. After that, we establish the theoretical guarantees on the downstream performance of the features before the projector, which characterize the role of the projection head in contrastive learning and indicate the principle for designing an effective projector.

### 3.1 PROBLEM SETUPS

**Information-theoretic Formulation.** As shown in Figure 1, the overall information flow in contrastive learning transmits from $Y$ to $R$. To elaborate, $Y$ represents the natural semantics that generates input samples $X$ (e.g., the supervised labels), $Z_1$ and $Z_2$ are the encoder features and projector features, and $R$ are the self-supervised targets in contrastive learning (positive samples in contrastive learning are pulled together while negative samples are pushed away). Following HaoChen et al. (2021), we assume that the set of $X$ is a finite but exponentially large set for the simplicity of analysis. During the contrastive pretraining process, the backbone network encodes the input samples and obtains the encoder features $Z_1$. After that, the projector is applied following the encoder and we obtain the projector features $Z_2$. Eventually, we calculate the contrastive objective on the projector features. **Consequently, our information flow model can be summed up as** $Y \rightarrow X \rightarrow Z_1 \rightarrow Z_2 \rightarrow R$**.**

**Mutual Information Definitions.** We then introduce the basic concept of mutual information. We adopt Shannon information, where the entropy of a random variable $X$ is defined as $H(X) = -\mathbb{E}_{P(X)} \log P(X)$, and the mutual information between two variables $X_1$ and $X_2$ is $I(X_1; X_2) = H(X_1) - H(X_1|X_2) = H(X_2) - H(X_2|X_1)$. Intuitively, $I(X_1; X_2)$ conveys the reduction in the entropy of $X_1$ when the information about $X_2$ is given. When $I(X_1; X_2) = 0$, $X_1$ and $X_2$ are independent variables, and when they are closely related, their mutual information tends to be large.

**Mutual Information Dynamics in Contrastive Learning.** We now utilize the concept of mutual information to revisit the contrastive learning process. During pretraining, we train the projector feature with infoNCE loss, aligning similar objects while distancing dissimilar ones, which increases the mutual information $I(Z_2; R)$ (Oord et al., 2018). It has been proved that minimizing the contrastive objectives enables the neural network to obtain essential information for downstream tasks (e.g., the supervised labels in classification tasks), i.e., $I(Z_2; Y)$ increases with the larger $I(Z_2; R)$ (Tian et al., 2020). However, after pretraining is completed, the common contrastive pipeline discards the projection head and uses the encoder feature $Z_1$ for downstream tasks, which suggests that we should focus on establishing the theoretical guarantees of $I(Z_1; Y)$.

### 3.2 Theoretical Guarantees of Downstream Task Performance

In this section, we will first theoretically derive a lower bound of $I(Y; Z_1)$, which provides a new understanding of how pretraining the projection head can benefit downstream tasks.

**Theorem 3.1.** *The downstream task performance of encoder features can be lower-bounded by*
$$I(Y; Z_1) \geq I(Z_1; R) - I(Z_1; Z_2) + I(R; Y).$$

*Proof Sketch.* We provide a proof sketch of this lower bound as it is the core of our theoretical analysis. We first utilize the unidirectional information flow model to illustrate that $Y$ and $R$ are conditionally independent when given the encoder feature, i.e., $I(Y; R|Z_1) = 0$. Besides, we observe that $Z_1$ learns less from $R$ than from $Z_2$, i.e., $I(Z_1; R) \leq I(Z_1; Z_2)$. With these two observations, we derive the lower bound.

$$
\begin{aligned}
I(Y; Z_1) &= I(Y; Z_1) + I(Y; R|Z_1) && (I(Y; R|Z_1) = 0) \\
&= I(Y; Z_1, R) \\
&= H(Z_1; R) - H(Z_1, R|Y) \\
&\geq H(Z_1, Z_2) - H(Z_1, R|Y) + H(R) - H(Z_2) && (I(Z_1; R) \leq I(Z_1; Z_2)) \\
&= H(Z_1, Z_2) - H(Z_1|R, Y) - H(R|Y) + H(R) - H(Z_2) \\
&\geq H(Z_1, Z_2) - H(Z_1|R) - H(R|Y) + H(R) - H(Z_2) && (H(Z_1|R) \geq H(Z_1|R, Y)) \\
&= H(Z_1|Z_2) - H(Z_1|R) + I(R; Y) \\
&= I(Z_1; R) - I(Z_1; Z_2) + I(R; Y).
\end{aligned}
$$

According to Theorem 3.1, the lower bound comprises three components. As the mutual information between the ground-truth labels and self-supervised targets ($I(R; Y)$) can be regarded as a constant in contrastive learning (Arora et al., 2019), we focus on the first two terms, i.e., $I(Z_1; R)$ and $I(Z_1; Z_2)$.

**The Projector Needs to Preserve Sufficient Information Related to Contrastive Objectives.** The first term, $I(Z_1; R)$, measures the amount of information that the encoder feature acquires from the contrastive learning objective. To improve the lower bound of downstream task performance, the encoder feature need to share sufficient knowledge of the self-supervised target $R$. As the contrastive objective is calculated on the projector features, this term implies that the projector should avoid forgetting the essential information, which is consistent with the practical design where the projector is usually a shallow architecture.

**The Projector Needs to Filter out Unrelated Information.** As for the second term, the theorem states that the downstream performance improves with less mutual information between encoder and projector features. The result indicates the necessity of the projection head. Without the projector, this term achieves the maximum, which indicates the downstream performance significantly declines. In practice, several studies improve the downstream performance of contrastive learning by implicitly decreasing the mutual information between encoder and projector features. For example, Jing et al. (2021) discard part of the projector features when calculating the contrastive loss and Lavoie et al. (2022) project the encoder features into a simplicial representation space.

Combining two terms, the theorem delivers a principle for designing a projector: it should act as an information bottleneck (Tishby et al., 2000) that filters out the irrelevant information of the contrastive objective.

Furthermore, we also provide an upper bound of $I(Y; Z_1)$ to analyze the influence of the projection head.

**Theorem 3.2.** *The downstream task performance of encoder features can be upper bounded by*

$$I(Y; Z_1) \leq I(Y; Z_2) - I(Z_1; Z_2) + H(Z_1).$$

*Proof Sketch.* We provide a proof sketch of this upper bound. Analogous to the proof of the lower bound, we observe that $Y$ and $Z_2$ are conditionally independent wen given the encoder feature, i.e., $I(Y; Z_2|Z_1) = 0$. We then obtain the upper bound with some simple deduction.

$$
\begin{aligned}
I(Y; Z_1) &= I(Y; Z_1, Z_2) & (I(Y; Z_2|Z_1) = 0) \\
&= I(Y; Z_1|Z_2) + I(Y; Z_2) \\
&\leq H(Z_1|Z_2) + I(Y; Z_2) \\
&= I(Y; Z_2) + H(Z_1) - I(Z_1; Z_2).
\end{aligned}
$$

Theorem 3.2 indicates that the downstream performance gap between the encoder features and projector features is decided by two terms, i.e., $I(Z_1; Z_2)$ and $H(Z_1)$. Similar to the lower bound of the downstream performance, the first term $I(Z_1; Z_2)$ implies that an effective projector should decrease the mutual information between encoder and projector features. As for the second term ($H(Z_1)$), the bound implies that the encoder features with better downstream performance should be more uniform in the representation space, which is consistent with the empirical findings (Ma et al., 2023). The upper bound further verifies the principle for designing an effective projector, i.e., the projector should be able to filter information unrelated to contrastive objectives.

In the following, we first evaluate the effectiveness of the theoretical analysis with existing projectors and then propose several improvements based on the theoretical principle.

### 3.3 EMPIRICAL VERIFICATIONS ON THEORETICAL GUARANTEES

In this section, we will justify our theoretical guarantees by validation experiments. Due to the difficulty in computing mutual information with high precision, we adopt the matrix mutual information as a surrogate metric (Tan et al., 2023). We introduce their definitions of matrix entropy and matrix mutual information as follows.

**Definition 3.3** (Matrix-based $\alpha$-order (Rényi) entropy). Suppose matrix $A \in \mathbf{R}^{n \times n}$ which $A(i, i) = 1$ for every $i = 1, 2, ...n$. $\alpha$ is a positive real number. The $\alpha$-order (Rényi) entropy for matrix $A$ is defined as follows:

$$H_\alpha(A) = \frac{1}{1 - \alpha} \log \left[ \mathrm{tr} \left( \left( \frac{A}{n} \right)^\alpha \right) \right],$$

where we use $\alpha = 2$ as the default value.

**Definition 3.4** (Matrix-based mutual information). Suppose matrix $A, B \in \mathbf{R}^{n \times n}$ which $A(i, i) = B(i, i) = 1$ for every $i = 1, 2, ...n$. $\alpha$ is a positive real number. The $\alpha$-order (Rényi) mutual information between $A$ and $B$ is defined as follows:

$$I_\alpha(A; B) = H_\alpha(A) + H_\alpha(B) - H_\alpha(A \odot B),$$

where $\odot$ represents the Hadamard product of two matrices.

Following Tan et al. (2023), we respectively estimate the mutual information $I(Z_1; Z_2)$ and entropy $H(Z_1)$ with the matrix surrogate metric $I_\alpha(\hat{Z}_1 \hat{Z}_1^\top; \hat{Z}_2 \hat{Z}_2^\top)$ and $H_\alpha(\hat{Z}_1 \hat{Z}_1^\top)$, where $\hat{Z}_1, \hat{Z}_2$ are the normalized encoder and projector feature matrices of the samples. As for estimating $I(Z_1; R)$, we calculate the contrastive loss of the encoder features as a surrogate metric (Oord et al., 2018).

**Tendency during Training.** We start with observing the changing tendency of estimated theoretical bounds during the pretraining process of contrastive learning. In practice, we take the widely-used contrastive method SimCLR (Chen et al., 2020) as an example and train the ResNet-18 on CIFAR-100. As shown in Figure 2, both lower and upper bounds continuously increase during training. Meanwhile, the rate of increase is rapid initially and gradually slows down, which is consistent with

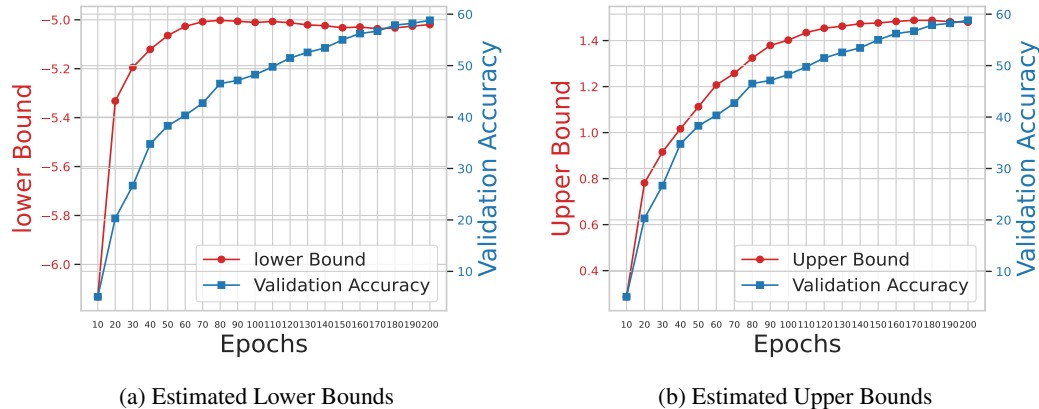

(a) Estimated Lower Bounds

(b) Estimated Upper Bounds

Figure 2: The change processes of estimated guarantees and practical downstream performance of encoder features trained with SimCLR on CIFAR-100 for 200 epochs. The trend indicates that the bounds provide a fairly accurate estimation of the variations in downstream performance.

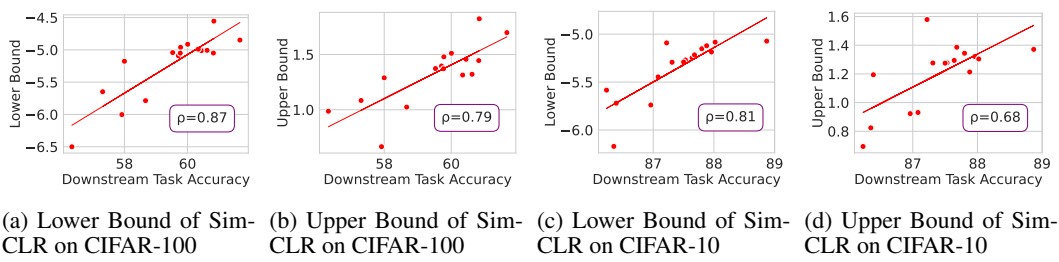

(a) Lower Bound of Sim-CLR on CIFAR-100

(b) Upper Bound of Sim-CLR on CIFAR-100

(c) Lower Bound of Sim-CLR on CIFAR-10

(d) Upper Bound of Sim-CLR on CIFAR-10

Figure 3: Correlation between downstream task accuracy and the estimated theoretical bounds on CIFAR-10 and CIFAR-100. Different points represent the encoder features learned by SimCLR with different projectors.

the tendency of the downstream task performance. The empirical results indicate that the estimated theoretical bounds accurately characterize the change process of the downstream performance with encoder features.

**Correlation between Bounds and Downstream Accuracy.** We then investigate the correlation between the theoretical guarantees and downstream accuracy with different projectors. Specifically, we compare projectors with different depths, different widths, different training parameters, and different implementations. As shown in Figure 3, both the estimated lower and upper bounds exhibit strong correlations to the practical downstream accuracy, which further verifies the effectiveness of our theoretical estimation on the downstream performance of encoder features. More empirical details can be found in Appendix B.1.

## 4 NEW PROJECTORS WITH PRINCIPLED MODIFICATIONS

The theoretical analysis in Section 3 indicates that an effective projection head should be an information bottleneck and filter out the information irrelevant to contrastive objectives. Built on this principle, we optimize the designs of projection heads through two approaches: training regularization and structural regularization. With modified projectors, we evaluate the performance of contrastive learning in various real-world datasets.

### 4.1 TRAINING REGULARIZATION

To control the mutual information between encoder and projector features, a straightforward method is to estimate the mutual information and apply it as a regularization term to the contrastive loss. As the accurate estimation of mutual information is not accessible, we still adopt the matrix mutual information introduced in Section 3.3 as the surrogate metric and denote it as the bottleneck regu-

Table 1: Linear evaluation accuracy (%) of contrastive learning representations learned by the objectives with/without regularizations on the mutual information between encoder and projector features.

| Dataset | Framework | Baseline | Bottleneck Regularizer | Gains |
|---|---|---|---|---|
| CIFAR-10 | SimCLR | 87.47 | 87.73 | +0.26 |
| | Barlow Twins | 89.00 | 89.32 | +0.32 |
| CIFAR-100 | SimCLR | 58.12 | 59.38 | +1.26 |
| | Barlow Twins | 64.16 | 66.31 | +2.15 |
| ImageNet-100 | SimCLR | 67.36 | 68.42 | +1.06 |
| | Barlow Twins | 69.96 | 71.3 | +1.34 |

larizer:

$$\mathcal{L}_{reg} = I_\alpha(\hat{Z}_1 \hat{Z}_1^\top ; \hat{Z}_2 \hat{Z}_2^\top). \tag{1}$$

Taking the widely-used InfoNCE loss as an example, the modified objective is:

$$\mathcal{L}_{total} = -\mathbb{E}_{x,x^+,\{x^-\}_{i=1}^n} \log \frac{\exp(f(x)^\top f(x^+))}{\exp(f(x)^\top f(x^+)) + \sum_{i=1}^n \exp(f(x)^\top f(x^-))} + \lambda \cdot \mathcal{L}_{reg}, \tag{2}$$

where $(x, x^+)$, $(x, x^-)$ are the positive and negative pairs in contrastive learning, $\lambda$ is a hyper-parameter to control the scale of regularization.

**Setup.** We conduct our experiments on CIFAR-10, CIFAR-100, and ImageNet-100, with Resnet-18 as our backbone. We evaluate the classification based on two representative contrastive frameworks, i.e., SimCLR (Chen et al., 2020) and Barlow Twins (Zbontar et al., 2021). On CIFAR-10 and CIFAR-100, we train the model for 200 epochs with batch size 256 and weight decay $10^{-4}$. The coefficient of the regularization term ($\lambda$) is set to 0.0001. On ImageNet-100, we train the model for 100 epochs with batch size 128 and weight decay $10^{-4}$. The coefficient of the regularization term is set to 0.01. More details can be found in Appendix B.2.

**Empirical Results.** In Table 1, we compare the performance of the original InfoNCE loss with ours across different benchmarks. Under the SimCLR framework, we find that the classification accuracy increases by 0.26% on CIFAR-10, 1.26% on CIFAR-100, and 1.06% on ImageNet-100 with the regularization term. Meanwhile, under the Barlow Twins framework, the classification accuracy increases by 0.32% on CIFAR-10, 2.15% on CIFAR-100, and 1.34% on ImageNet-100. The experimental results consistently suggest that adding the bottleneck regularizer to control the mutual information between encoder and projector features can effectively improve the downstream performance across different datasets and contrastive frameworks.

## 4.2 STRUCTURAL REGULARIZATION

In addition to applying a regularizer to training objectives, we also explore controlling the mutual information between encoder and projector features by modifying the projector structures. Specifically, we use two common structures that can control the amount of preserved information: feature discretization (Liu et al., 2002) and sparse autoencoder (Ng et al., 2011).

### 4.2.1 DISCRETIZED PROJECTOR

As a well-known technique to remove irrelevant information (Kotsiantis & Kanellopoulos, 2006), discretization converts continuous features to discrete points in the representation space, where the number of discrete points determines how much information is retained. For simplicity, we follow the implementation of discretization proposed by Mentzer et al. (2023) and adopt the Finite Scalar Quantization structure (FSQ). Specifically, we discrete each dimension of projector features as follows:

$$\hat{f}(x)_i = \lfloor g(f(x)_i) + \frac{1}{2} \rfloor, \tag{3}$$

where $f(x)_i$ is the $i$-th dimension of the original projector feature, $g(z) = \lfloor L/2 \rfloor \tanh(z)$. We note that the tanh function enables the projector features to satisfy that $\tanh(z_i) \in (-1, 1)$. Consequently, $L$ decides the number of points in the discretization space, i.e., the mutual information between the encoder and projector features is controlled by $L$. When $L$ decreases, more information from the encoder features is discarded.

Table 2: Linear evaluation accuracy (%) of contrastive learning methods with the original and discrete projectors on CIFAR-10, CIFAR-100 and ImageNet-100.

| Dataset | Framework | Baseline | Discretized Projector | Gains |
|---|---|---|---|---|
| CIFAR-10 | SimCLR | 87.47 | 88.5 | +1.03 |
| | Barlow Twins | 89.00 | 89.38 | +0.38 |
| CIFAR-100 | SimCLR | 58.12 | 60.16 | +2.04 |
| | Barlow Twins | 64.16 | 65.18 | +1.02 |
| ImageNet-100 | SimCLR | 67.36 | 68.38 | +1.02 |
| | Barlow Twins | 69.96 | 71.68 | +1.72 |

**Setup.** Similar to the training regularization setting, we adopt SimCLR (Chen et al., 2020) and Barlow Twins (Zbontar et al., 2021) as our baseline frameworks, and measure the results on CIFAR-10, CIFAR-100 and ImageNet-100. The parameters are consistent with those of the training regularization setting. For the discretized projector, we discrete each dimension of the projector feature to 30 points in SimCLR and 3 points in Barlow Twins. More details can be found in Appendix B.3.

**Empirical Results.** We compare the performance of the original projectors with our discretized ones across different benchmarks. As displayed in Table 2, the classification accuracy increases by 1.03% on CIFAR-10, 2.04% on CIFAR-100, and 1.02% on ImageNet-100 with the discretized projector under the SimCLR framework. Meanwhile, under the Barlow Twins framework, the classification accuracy increases by 0.38% on CIFAR-10, 1.02% on CIFAR-100, and 1.71% on ImageNet-100. The experimental results on these benchmarks indicate that employing the discretized projector can promote downstream performance across different datasets and contrastive frameworks.

**Theoretical Understanding.** Beyond empirical improvements, we also provide a theoretical understanding on the gains of discretized projectors. The following theorem demonstrates that the discretized projector indicates superior downstream performance of encoder features. The proofs can be found in Appendix C.

**Theorem 4.1.** *The downstream performance of encoder features can be lower-bounded by*

$$I(Y; Z_1) \geq -H(Z_2) + I(Z_2; R) + I(R; Y). \tag{4}$$

The above theorem gives a theoretical explanation for the discretized projector. Generally, the constrastive loss (*i.e.,* $I(Z_2; R)$) is fairly low and can be regarded as a constant. Meanwhile, $I(R; Y)$ is also a constant. Thus, in order to increase the lower bound, the projector feature needs to be more concentrated, so that $H(Z_2)$ becomes smaller. Intuitively, discretizing the projector feature to several points can achieve this and greatly benefit downstream performance. Meanwhile, the existence of sweet point is well explained by the theorem. On the one hand, if the point number is too few (*e.g.,* one single point), then the contrastive learning process will collapse (*i.e.,* $I(Z_2; R)$ becomes very small since $Z_2$ can not acquire adequate information from $R$). On the other hand, if we do not discretize the projector feature, then $Z_2$ will become uniform (*i.e.,* $H(Z_2)$ is a very large value), reducing the lower bound dramatically. The analysis above provides a theoretical perspective to understand why there is a peak when the point number grows from one to infinity.

### 4.2.2 SPARSE PROJECTOR

Besides the discretization, we explore applying another powerful method for filtering our irrelevant information: the sparse autoencoder (Ng et al., 2011), which reconstructs the original features from a sparsely activated bottleneck layer. To be specific, we use the top-$k$ autoencoder (Gao et al., 2024) as:

$$\begin{aligned} h &= \text{Topk}(W_{\text{enc}}(f(x) - b_{\text{pre}})), \\ z_2 &= W_{\text{dec}}h + b_{\text{pre}}, \end{aligned} \tag{5}$$

where $W_{\text{enc}}$, $b_{\text{pre}}$, and $W_{\text{dec}}$ are trainable parameters, TopK is an activation function that only keeps the $k$ largest hidden representations. Serving as an information bottleneck, this approach decreases the mutual information between encoder and projector features when $k$ becomes smaller. Meanwhile, we figure out that the hidden layer only preserves the information of the top-activated features, which helps to retain the crucial aspects of encoder features, i.e., the information related to the contrastive objectives. Based on the theoretical analysis in Section 3, we believe that the sparse

Table 3: Linear evaluation accuracy (%) of contrastive learning methods with the original and sparse projectors on CIFAR-10, CIFAR-100, and ImageNet-100.

| Dataset | Framework | Baseline | Sparse Projector | Gains |
|---------|-----------|----------|------------------|-------|
| CIFAR-10 | SimCLR | 87.47 | 88.16 | +0.69 |
| | Barlow Twins | 89.00 | 89.48 | +0.48 |
| CIFAR-100 | SimCLR | 58.12 | 61.99 | +3.87 |
| | Barlow Twins | 64.16 | 68.15 | +3.99 |
| ImageNet-100 | SimCLR | 67.36 | 68.48 | +1.12 |
| | Barlow Twins | 69.96 | 71.86 | +1.90 |

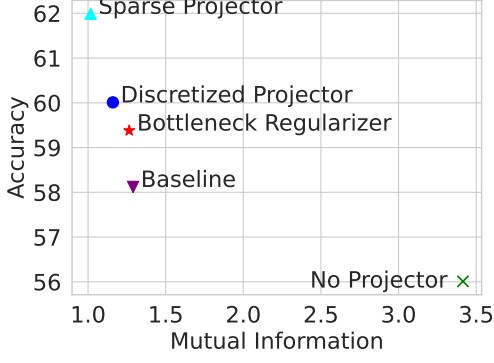

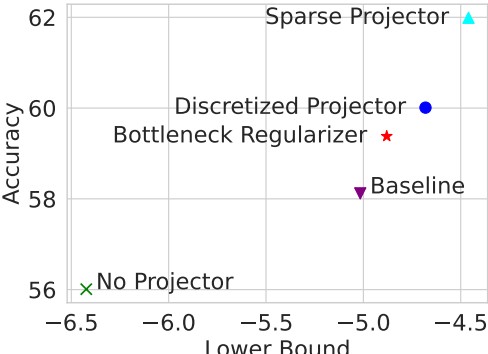

(a) Mutual Information between Encoder and Projector Features

(b) Estimated Lower Bounds of Downstream Performance

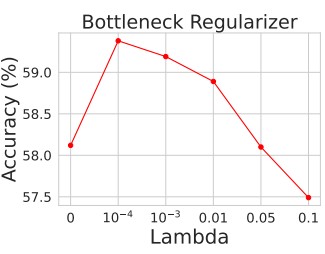

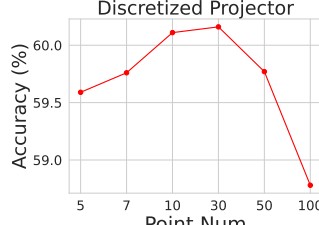

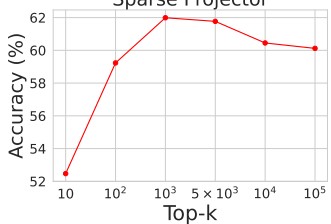

(c) Scales of Training Regularizers

(d) Numbers of Discrete Points

(e) Numbers of Activated Features

Figure 4: Empirical understandings of proposed methods. (a), (b) show the correlation between estimated theoretical guarantees and downstream performance. The proposed methods improve the downstream performance by filtering out the irrelevant information, leading to a better downstream performance guarantee. (c), (d), (e) demonstrate the influence of different regularization parameters. With stronger regularizations, the downstream performance increases first and then decreases.The experiments are conducted on the models trained by SimCLR on CIFAR-100.

autoencoder is an effective projector as we can filter out irrelevant semantics and only preserve the information related to the contrastive objectives.

**Setup.** The datasets and most parameters in this section are the same as those of the discretized projector. For the sparse projectors, we set aggressive ratios of unactivated neurons. Denote the dimension of the hidden layer $h$ in the sparse autoencoder as $d$, we set $k = 0.001d$ for SimCLR on CIFAR-10, CIFAR-100, and ImageNet-100. For Barlow Twins, we set $k = 0.001d$ on CIFAR10 and $k = 0.2d$ on CIFAR-100 and ImageNet-100. More details can be found in Appendix B.4.

**Empirical Results.** We compare the performance of the original projectors with our sparse ones across different benchmarks. As shown in Table 3, under the SimCLR framework, the sparse projector achieves an increase in classification accuracy by 0.69% on CIFAR-10, 3.87% on CIFAR-100, and 1.12% on ImageNet-100. Under the Barlow Twins framework, it also manages to achieve an increase in classification accuracy by 0.48% on CIFAR-10, 3.99% on CIFAR-100, and 1.90% on

ImageNet-100. The outcomes consistently suggest that the sparse projector can successfully improve the downstream performance across different datasets and contrastive learning frameworks.

### 4.3 Empirical Understandings of Proposed Methods

In previous sections, we modify the projectors by limiting the mutual information between encoder and projector features with training and structural regularizations. The empirical results show that our methods significantly improve the performance across different real-world datasets and contrastive frameworks.

In this section, we aim to establish a further understanding on the relationship between the empirical benefits and theoretical principles. The experiment details can be found in Appendix B.5.

**Validation of Proposed Methods.** We begin by analyzing the relationship between the impact of our proposed methods on theoretical guarantees and on practical performance. With the features trained by the SimCLR with no projector, the default projectors and our proposed projectors, we collect the downstream accuracy, the estimated lower bound and the estimated mutual information. In Figure 4a and 4b, we observe that our methods reduce the mutual information $I(Z_1; Z_2)$ and obtain a higher theoretical estimation of downstream performance. Furthermore, the degree of practical improvements strongly correlates with the improvements in theoretical guarantees. These results further validate the effectiveness of both our theoretical principles and proposed modifications.

**Ablation Study of Regularization Strengths.** We then conduct ablation study on the effect of different parameters in our methods. To be specific, we respectively vary the coefficient of the bottleneck regularization term, the point number of discretization, and the activation number of the sparse projector for the three methods. The results are displayed in Figure 4c, 4d, and 4e. For all of three methods, with the increase of discarded information by the regularized projectors, the downstream performance increases first and then decreases. The empirical results indicate that the original projector preserves too much information irrelevant to the contrastive objective and we can find the sweet point by gradually enhance the strength of regularizations.

## 5 Conclusion

Among various special designs in contrastive learning, the projection head is a crucial component contributing to its impressive performance across various downstream tasks. However, the underlying mechanism behind the projection head is still under-explored. In this paper, we present a new theoretical understanding of the projection head from an information-theoretic perspective. Particularly, by establishing the lower and upper bounds of the downstream performance, we reveal that an effective projector should act as an information bottleneck and filter out information irrelevant to contrastive objectives. Based on the theoretical analysis, we propose training and structural regularization methods to control the mutual information between encoder and projector features. Empirically, the proposed methods achieve consistent improvement on the downstream task across different datasets and contrastive frameworks. With the fruitful insights presented in this paper, we believe it has great potential and can inspire more principled designs in contrastive learning.

## Reproducibility Statement

To ensure the reproducibility of this paper, we detail the theoretical and empirical parts of our results in the main paper and the appendix. In Section 3 of the main paper, we introduce the basic notaions and models used in the theoretical analysis. Furthermore, in Appendix A, we provide the detailed proofs of the theoretical guarantees. For empirical details, we briefly introduce the modified projectors in Section 4 of the main paper. In Appendix B, we elaborate the parameters of our modified projectors, including the bottleneck regularizer, the discrtized projector and the sparse projector, and the settings where we pretrain and evaluate the models.

## Acknowledgement

Yisen Wang was supported by National Key R&D Program of China (2022ZD0160300), National Natural Science Foundation of China (92370129, 62376010), and Beijing Nova Program (20230484344, 20240484642). Yifei Wang was supported in part by the NSF AI Institute TILOS, and an Alexander von Humboldt Professorship.

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

## A OMITTED PROOFS

### A.1 AUXILIARY LEMMAS

To prove the lower and upper bounds, we first introduce some basic lemmas.

**Lemma A.1** (Information cutoff theory). *With information flow $Y \to X \to Z_1 \to Z_2 \to R$, we have $I(Y; R|Z_1) = 0$ and $I(Y; Z_2|Z_1) = 0$.*

*Proof.* From our model, we can obtain a simplified information chain: $Y \to Z_1 \to R$. Given $Z_1$, the information channel between $Y$ and $R$ is blocked out, i.e.,

$$P(R|Y, Z_1) = P(R|Z_1),$$

where $P(\cdot)$ denotes the probabilities. Consequently, we have

$$P(R, Y|Z_1) = P(Y|Z_1) \cdot P(R|Y, Z_1) = P(Y|Z_1) \cdot P(R|Z_1),$$

where the first equation is the Chain Rule of Conditional Probability. According to the definition of conditional mutual information, we have

$$I(Y; R|Z_1) = \sum_{y \in \mathcal{Y}} \sum_{r \in \mathcal{R}} \sum_{z_1 \in \mathcal{Z}_1} p(y, r, z_1) log \frac{p(y, r|z_1)}{p(r|z_1)p(y|z_1)} = 0.$$

The second equation in this lemma can be proved by the same technique. $\square$

**Lemma A.2** (Information processing theory). *With information flow $Z_1 \to Z_2 \to R$ and the information processing inequality, we have*

$$I(Z_1; Z_2) \geq I(Z_1; R) \quad and \quad I(Z_2, R) \geq I(Z_1; R).$$

*Proof.* With information flow $Z_1 \to Z_2 \to R$ and lemma A.1, we have $I(Z_1; R|Z_2) = 0$ and $I(Z_2; R|Z_1) = 0$. Consequently, the following two inequalities holds:

$$I(Z_1; Z_2) = I(Z_1; Z_2) + I(Z_1; R|Z_2) = I(Z_1; R, Z_2) \geq I(Z_1; R),$$
$$I(R; Z_2) = I(R; Z_2) + I(R; Z_1|Z_2) = I(R; Z_1, Z_2) \geq I(R; Z_1).$$

$\square$

### A.2 PROOF OF THEOREM 3.1

*Proof.* With lemma A.1, we know that $I(Y; R|Z_1) = 0$, which yields
$$I(Y; Z_1) = I(Y; Z_1) + I(Y; R|Z_1) = I(Y; Z_1, R).$$
Additionally, lemma A.2 claims that $I(Z_1; R) \leq I(Z_1; Z_2)$, which further gives

$$
\begin{aligned}
H(Z_1, R) &= H(Z_1) + H(R) - I(Z_1; R) \\
&\geq H(Z_1) + H(R) - I(Z_1; Z_2) \\
&= H(Z_1; Z_2) - H(Z_2) + H(R).
\end{aligned}
\tag{6}
$$

Thus, we can deduce the lower bound as follows:

$$
\begin{aligned}
I(Y; Z_1) &= I(Y; Z_1, R) \\
&= H(Z_1, R) - H(Z_1, R|Y) \\
&\geq H(Z_1; Z_2) - H(Z_2) + H(R) - H(Z_1, R|Y) \\
&= H(Z_1; Z_2) - H(Z_2) + H(R) - H(R|Y) - H(Z_1|R, Y) \\
&\geq H(Z_1|Z_2) - H(Z_1|R) + I(R; Y) \\
&= I(Z_1; R) - I(Z_1; Z_2) + I(R; Y).
\end{aligned}
$$

$\square$

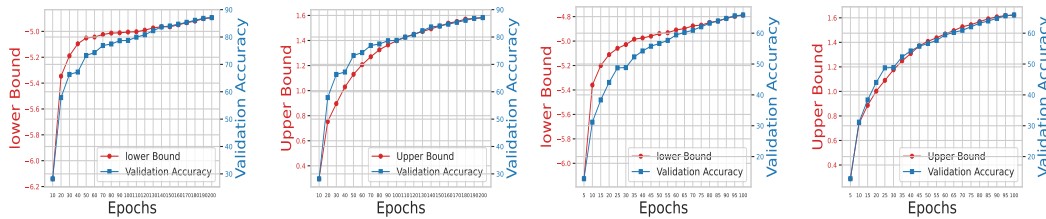

(a) Lower Bound Tendency During Training on CIFAR-10

(b) Upper Bound Tendency During Training on CIFAR-10

(c) Lower Bound Tendency During Training on ImageNet-100

(d) Upper Bound Tendency During Training on ImageNet-100

Figure 5: Tendency during training of SimCLR on CIFAR-10 and ImageNet-100.

### A.3 PROOF OF THEOREM 3.2

*Proof.* With lemma A.1, we know that $I(Y; Z_2|Z_1) = 0$, which yields

$$I(Y; Z_1) = I(Y; Z_1) + I(Y; Z_2|Z_1) = I(Y; Z_1, Z_2)$$

Then we can deduce the upper bound as follows:

$$
\begin{aligned}
I(Y; Z_1) &= I(Y; Z_1, Z_2) \\
&= I(Y; Z_1|Z_2) + I(Y; Z_2) \\
&\leq H(Z_1|Z_2) + I(Y; Z_2) \\
&= I(Y; Z_2) + H(Z_1) - I(Z_1; Z_2).
\end{aligned}
\tag{7}
$$

$\square$

## B EXPERIMENT DETAILS

In this section, we detail the setting of each individual experiment in this work. All experiments are conducted with at most two NVIDIA RTX 3090 GPUs.

### B.1 EXPERIMENT DETAILS OF EMPIRICAL VERIFICATION ON THEORETICAL GUARANTEES

In this part, we justify our theoretical guarantees by two kinds of validation experiments. On the one hand, we manage to verify that the tendency of bounds fits quite well with that of validation accuracy. On the other hand, we demonstrate that both bounds are highly correlated with downstream task performance.

**Tendency during training.** In this experiment, we aim to show the tendency of the lower and upper bounds during the pretraining process and then compare them with the trend of validation accuracy. We conduct the experiment under the SimCLR framework on CIFAR-100 with a total run of 200 epochs, using ResNet-18 as the backbone and an MLP as our projector. As for the hyper-parameters in detail, we use learning rate 0.4, weight decay $10^{-4}$, InfoNCE temperature 0.2. Then we record both bounds and validation accuracy every 10 epochs and plot the first 100 epochs.

As a supplemental aspect for the experiments of tendency during training, we plot the tendency of baselines from SimCLR and Barlow Twins on CIFAR-10, CIFAR-100, and ImageNet-100. To be more specific, Figure 2a, 5 and 6 respectively represent the baseline of SimCLR on CIFAR-100, the baselines of SimCLR on CIFAR-10 and ImageNet-100, and the baselines of Barlow Twins on CIFAR-100, CIFAR-10, ImageNet-100.

**Correlation between bounds and downstream accuracy.** We investigate the correlation between the upper and lower bounds and downstream task accuracy with different projectors. In these experiments, we utilize the SimCLR framework on both CIFAR-10 and CIFAR-100 with ResNet-18 as the backbone and train for 200 epochs using batch size 256. As for the hyper-parameters, we use learning rate 0.4, weight decay $10^{-4}$, InfoNCE temperature 0.2, augmentation cropsize 32.

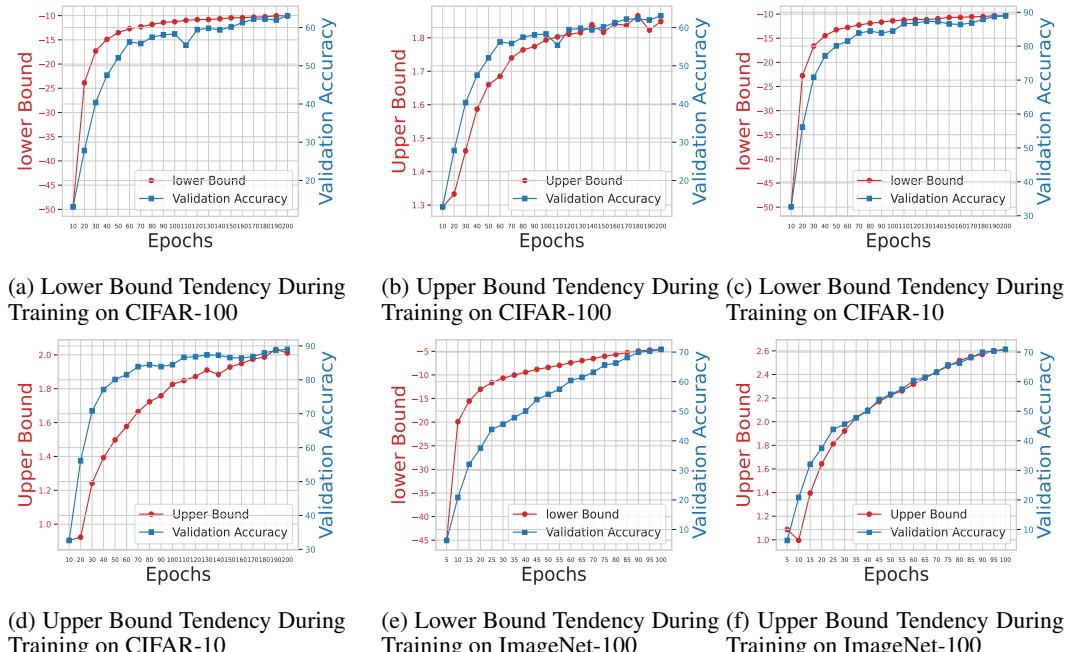

Figure 6: Tendency during training of Barlow Twins on CIFAR-10, CIFAR-100 and ImageNet-100.

To be more specific, we adopt linear projectors with different output dimensions (128, 256, 512, 1024), MLP projectors (i.e., Linear-ReLU-Linear) with different output dimensions (128, 256, 512, 1024), no projector (i.e., use the encoder feature to calculate the contrastive loss), a deepened-MLP projector (Linear-ReLU-Linear-ReLU-Linear) and two improved projectors (identity-mapping and DirectCLR) on both CIFAR-10 and CIFAR-100. As replenishing experiments, we change learning rate to 0.2, weight decay to $10^{-3}$, InfoNCE temperature to 0.1, augmentation crop size to 64 respectively on both CIFAR-10 and CIFAR-100. In total, these choices of parameters correspond to 16 experiments and 16 dots in Figure 3. We calculate the covariance between the lower and upper bounds and downstream task accuracy, using it as a criterion to validate the effectiveness of our theoretical estimation on the downstream performance of encoder features.

As a supplemental aspect for the experiments of correlation, we also investigate the bounds and downstream performance of different projectors of SimCLR on ImageNet-100 and Barlow Twins on CIFAR-100. We plot the results in Figure 7, in which different points represent the experiments conducted with different projectors. To be more specific, for SimCLR on ImageNet-100, we adopt linear projectors with different output dimensions (128, 256, 512, 1024), MLP projectors (i.e., Linear-ReLU-Linear) with different output dimensions (128, 256, 512, 1024), no projector (i.e., use the encoder feature to calculate the contrastive loss). For Barlow Twins on CIFAR-100, we adopt full Barlow Twin's projectors with different hidden dimensions (1024, 4096) and different ouput dimensions (1024, 4096), half of Barlow's projectors (i.e., Linear-BN-ReLU-Linear) with different hidden dimensions (1024, 4096), output dimensions (1024, 4096), baseline (with hidden dimension 2048 and output dimension 2048) and no projector.

## B.2 EXPERIMENT DETAILS OF TRAINING REGULARIZATION

In this experiment, we add a regularization term to the loss function to optimize the training of the projector. We conduct this experiment on CIFAR-10, CIFAR-100 and ImageNet-100, using SimCLR and Barlow Twins as our frameworks. For all experiments, we adopt ResNet-18 as the backbone and use an MLP projector whose structure is dependent on the framework and will be introduced in the following. During the pretraining process, we simultaneously train a classifier, which is a single-layer linear head and is trained without influencing on the rest of the network.

Recall Definition 3.4, and that the regularization term is $\mathcal{L}_{reg} = I_\alpha(\hat{Z}_1\hat{Z}_1^\top; \hat{Z}_2\hat{Z}_2^\top)$, where we take $\alpha = 2$ as the default value. As for the original loss functions of SimCLR and Barlow Twins, we

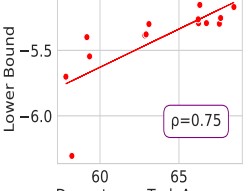 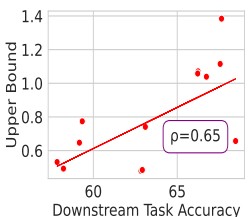 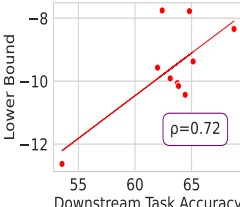 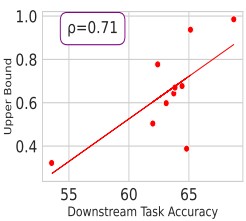

(a) Lower Bound of Sim-CLR on ImageNet-100    (b) Upper bound of Sim-CLR on ImageNet-100    (c) Lower Bound of Barlow Twins on CIFAR-100    (d) Upper Bound of Barlow Twins on CIFAR-100

Figure 7: Correlation between downstream task accuracy and the estimated theoretical bounds of SimCLR on ImageNet-100 and Barlow-Twins on CIFAR-100.

have

$$\mathcal{L}_{simclr} = -\mathbb{E}_{x,x^+,\{x^-\}_{i=1}^n} \log \frac{\exp(f(x)^\top f(x^+))}{\exp(f(x)^\top f(x^+)) + \sum_{i=1}^n \exp(f(x)^\top f(x^-))},$$

and

$$\mathcal{L}_{BT} = \sum_i (1 - \mathcal{C}_{ii})^2 + \gamma \sum_i \sum_{j \neq i} \mathcal{C}_{ij}^2$$

where $\mathcal{C}_{ij} = \frac{\sum_b z_{b,i}^A z_{b,j}^B}{\sqrt{\sum_b (z_{b,i}^A)^2} \sqrt{\sum_b (z_{b,j}^B)^2}}$, $z^A$ and $z^B$ denote the projector features of two views, $b$ is the batch size and $\gamma$ is a scaling factor. We then produce our loss functions by adding the regularization term, i.e.,

$$\mathcal{L}_{simclr\_total} = \mathcal{L}_{simclr} + \lambda \mathcal{L}_{reg} \quad \text{and} \quad \mathcal{L}_{BT\_total} = \mathcal{L}_{BT} + \lambda \mathcal{L}_{reg}.$$

For experiments conducted on CIFAR-10 and CIFAR-100, we use batch size 256 and train for 200 epochs, while for those conducted on ImageNet-100, we use batch size 128 and train for 100 epochs.

**SimCLR.** We detail other hyper-parameters in the SimCLR framework. On CIFAR-10 and CIFAR-100, we use learning rate 0.4, weight decay $10^{-4}$, InfoNCE temperature 0.2, and set $\lambda$ to $10^{-4}$. Our projector adopts a Linear-ReLU-Linear structure, where we use 2048 as the hidden dimension and 256 as the output dimension. On ImageNet-100, we use learning rate 0.3, weight decay $10^{-4}$, InfoNCE temperature 0.2 , and set $\lambda$ to 0.01. We use the same projector structure but change the hidden dimension to 4096 and the output dimension to 512.

**Barlow Twins.** We detail other hyper-parameters in the Barlow Twins framework. On CIFAR-10 and CIFAR-100, we use learning rate 0.3, weight decay $10^{-4}$, scaling factor $5 \times 10^{-3}$, and set $\lambda$ to $10^{-3}$. Our projector consists of three linear layers separated by two pairs of batch normalization and ReLU layers, where we use 2048 as the hidden dimension and the output dimension. On ImageNet-100, we use learning rate 0.3, weight decay $10^{-4}$, scaling factor $5 \times 10^{-3}$, and set $\lambda$ to 0.01. We use the same projector structure as the one on CIFAR datasets.

### B.3 EXPERIMENT DETAILS OF DISCRETIZED PROJECTOR

In this experiment, we discretize each dimension of the projector to a certain number of points, which we denote as $L$. We conduct this experiment on CIFAR-10, CIFAR-100 and ImageNet-100, using SimCLR and Barlow Twins as our frameworks. All the network settings are exactly the same as those in Appendix B.2. For experiments conducted on CIFAR-10 and CIFAR-100, we use batch size 256 and train for 200 epochs, while for those conducted on ImageNet-100, we use batch size 128 and train for 100 epochs. We respectively take $L = 30$ under the SimCLR framework and $L = 3$ under the Barlow Twins framework.

### B.4 EXPERIMENT DETAILS OF SPARSE PROJECTOR

In this experiment, we employ the sparse autoencoder in our projection head, adding sparsity to the projector feature so as to reduce the similarity between encoder and projector features (i.e., decrease $I(Z_1; Z_2)$). To be specific, let us recall the formular of top-k autoencoder,

$$h = \text{Topk}(W_{\text{enc}}(f(x) - b_{\text{pre}})),$$

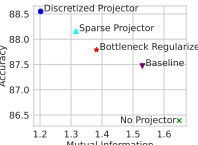
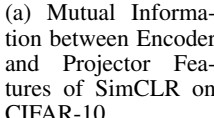
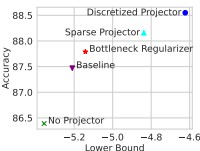
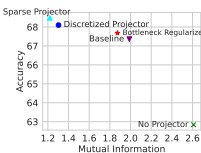
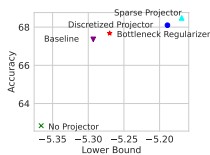

| (a) Mutual Information between Encoder and Projector Features of SimCLR on CIFAR-10 | (b) Estimated Lower Bounds of Downstream Performance of SimCLR on CIFAR-10 | (c) Mutual Information between Encoder and Projector Features of SimCLR on ImageNet-100 | (d) Estimated Lower Bounds of Downstream Performance of SimCLR on ImageNet-100 |
|---|---|---|---|

Figure 8: Supplemental experiments on the correlation between estimated theoretical guarantees and downstream performance of SimCLR on CIFAR-10 and ImageNet-100.

in which TopK is an activation function that only maintains the $k$ largest hidden representations and keeps the rest inactivated (i.e., set them to 0). We conduct this experiment on CIFAR-10, CIFAR-100 and ImagNet-100, utilizing SimCLR and Barlow Twins as our frameworks. For all experiments, we adopt ResNet-18 as the backbone, while the structure of the projector is dependent on the framework we choose and will be introduced in the following in detail. For experiments conducted on CIFAR-10 and CIFAR-100, we train for 200 epochs using batch size 256, while for those carried out on ImageNet-100, we train for 100 epochs with batch size 128.

**SimCLR.** Under the SimCLR framework, we directly employ the sparse autoencoder as our projector, totally deleting the original projector of SimCLR. Our projector adopts a structure similar to SimCLR (i.e., Linear-ReLU-Linear), but we activate only a tiny number of hidden representations. On CIFAR-10 and CIFAR-100, we use learning rate 0.4, weight decay $10^{-4}$, and InfoNCE temperature 0.2. To be specific, on CIFAR-10, we use hidden dimension $10^4$ and set $k = 100$ after trying a variety of different $k$. On CIFAR-100, we set the hidden dimension to $10^5$ and $k = 10^2$ after testing different $k$ from 10 to $10^5$. On ImageNet-100, we use learning rate 0.3, weight decay $10^{-4}$, InfoNCE temperature 0.2, fix the hidden dimension to $10^5$, and then set $k = 10^2$.

**Barlow Twins.** Under the Barlow Twins framework, we preserve half of its original projector and employ the sparse autoencoder following it, i.e., a Linear-BN-ReLU structure. On CIFAR-10 and CIFAR-100, we use learning rate 0.3, weight decay $10^{-4}$ and scaling factor $10^{-5}$. Similar to the SimCLR section, we use hidden dimension $10^5$ on CIFAR-10 and $10^6$ on CIFAR-100, and then set $k = 2 \times 10^3$ on CIFAR-10 and $k = 10^3$ on CIFAR-100. On ImageNet-100, we use learning rate 0.3, weight decay $10^{-4}$, scaling factor $10^{-5}$, hidden dimension $10^6$ and set $k = 2 \times 10^3$.

### B.5 EXPERIMENT DETAILS OF EMPIRICAL UNDERSTANDINGS OF PROPOSED METHODS

In this section, we use ResNet-18 as the backbone and SimCLR as our framework. During the pretraining process, we simultaneously train a classifier, which is a single-layer linear head and is trained without influencing on the rest of the network. As default parameters, we take learning rate 0.4, weight decay $10^{-4}$, and InfoNCE temperature 0.2, batch size 256, training epoch 200 on CIFAR-10 and CIFAR-100, and learning rate 0.3, weight decay $10^{-4}$, InfoNCE temperature 0.2, batch size 128, training epoch 100 on ImageNet-100. The projectors of our proposed methods follow the corresponding designs in previous sections.

**Validation of Proposed Methods.** In this part, we focus on the correlation between the estimated theoretical guarantees and downstream performance. The experiments are conducted on CIFAR-10, CIFAR-100 and ImageNet-100. We now detail the core parameters. On CIFAR-100, we set $\lambda = 10^{-3}$ for the bottleneck regularizer, $L = 20$ for the discretized projector and $k = 10^3$ for the sparse projector. On CIFAR-10, we set $\lambda = 10^{-4}$ for the bottleneck regularizer, $L = 20$ for the discretized projector and $k = 5 \times 10^3$ for the sparse projector. On ImageNet-100, we take $\lambda = 10^{-3}$ for the bottleneck regularizer, $L = 30$ for the discretized projector and $k = 10^3$ for the sparse projector. The result on CIFAR-100 is displayed in Figure 4a and 4b, while the rest can be found in Figure 8.

**Ablation Study of Regularization Strengths.** In this part, we study the effect of different parameters in our methods on CIFAR-100. For the bottleneck regularizer, we take $\lambda = 0, 10^{-4}, 10^{-3}, 0.01, 0.05, 0.1$ to form the line chart. As for the discretized projector, we

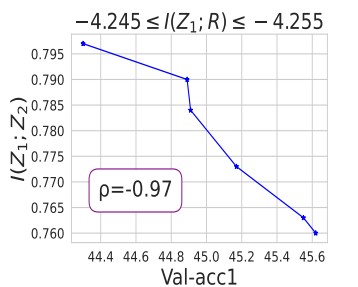 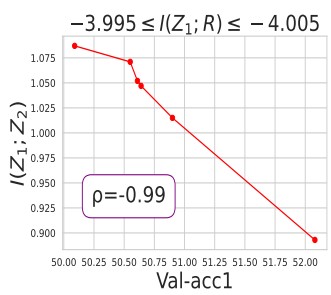 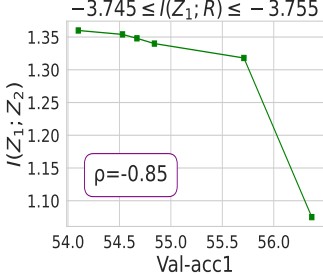

(a) Fix $I(Z_1; R)$ to around -4.250    (b) Fix $I(Z_1; R)$ to around -4.000   (c) Fix $I(Z_1; R)$ to around -3.750

Figure 9: Relationship between downstream performance and $I(Z_1; Z_2)$ with fixed $I(Z1; R)$.

take $L = 5, 7, 10, 30, 50, 100$. Regarding the sparse projector, we explore $k$ with values in $10, 100, 10^3, 5 \times 10^3, 10^4, 10^5$.

### B.6 CALCULATION OF MUTUAL INFORMATION

In all our experiments, we take $\alpha = 2$ when calculating mutual information. In this part, we clarify on the reason of our choice. Following (Tan et al., 2023), the 1-order (Rényi) entropy for matrix A where $\alpha = 1$ is defined as

$$H_1(A) = -\mathrm{tr}\left(\frac{A}{n} \log \frac{A}{n}\right).$$

Table 4: Linear probing accuracy of the models trained with bottleneck regularization (implemented by selecting different $\alpha$) on CIFAR-100.

| $\alpha$ | 1 | 2 | 3 |
|---|---|---|---|
| ACC | 59.06 | **59.38** | 58.59 |

This combined with Definition 3.3 yields the complete definition of matrix entropy and thus matrix mutual information. We then use the training regularization setting to test the downstream accuracy of SimCLR on CIFAR-100 with different choices of $\alpha$, where the coefficient of the regularization term is $10^{-4}$. In Table 4, we find that the downstream accuracy of the case with $\alpha = 2$ is slightly higher than that of the one with $\alpha = 1$, so $\alpha = 2$ tends to be a better choice in the sense of downstream performance. Moreover, we find that training with $\alpha = 1$ is way more time-consuming because of Singular Value Decomposition, which further motivates us to choose $\alpha = 2$ eventually.

### B.7 ABLATION STUDY OF INFORMATION BOTTLENECK $I(Z_1; Z_2)$

In order to rigorously verify that the downstream performance is indeed mainly influenced by $I(Z_1; Z_2)$, we collect data points with several fixed values of $I(Z_1; R)$ and study the relationship between downstream performance and $I(Z_1; Z_2)$. Typically, we choose three cases where $I(Z_1; R) \approx -4.25, -4.00, -3.75$ (with errors no more than 0.01) and collect 6 points for each case. The results in Figure 9 indicate that the downstream accuracy indeed rises as $I(Z_1; Z_2)$ falls, and the absolute value of correlation between $I(Z_1; Z_2)$ and the downstream accuracy when $I(Z_1; R)$ is fixed is close to 1, which further justifies the role of $I(Z_1; Z_2)$ as an information bottleneck.

### B.8 CORRELATION BETWEEN THE ONLINE ACCURACY OF ENCODER AND PROJECTOR FEATURES

In this part, we aim to investigate the correlation between $I(Y; Z_1)$ and $I(Y; Z_2)$. For this, we attach a new classifier after the projector to obtain the classification accuracy of the projector feature (i.e., $I(Y; Z_2)$). As shown in Figure 10, the correlation between the online accuracy of encoder and projector features is fairly small, indicating that $I(Y; Z_2)$ may not have a fixed relationship with

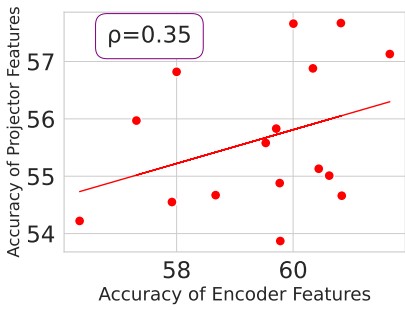

Figure 10: The correlation between the online accuracy of encoder and projector features.

$I(Y; Z_1)$. This further implies that although the inequality $I(Y; Z_1) \leq I(Y; Z_2)$ holds, maximizing $I(Y; Z_2)$ does not necessarily mean maximizing $I(Y; Z_1)$.

### B.9 COMPARISON BETWEEN TRAINING AND STRUCTURAL REGULARIZATION

Although both training and structural regularization can effectively improve downstream performance, we observe better results with structural regularization, which motivates us to explore the comparison between these two types of regularization. We conjecture that the underlying reason why training regularization has inferior downstream performance is that though it lowers the value of $I(Z_1; Z_2)$, it fails to protect $I(Z_1; R)$, resulting in a poor lower bound and thus poor downstream accuracy. To prove such an insight, we use larger coefficients for the regularization term to align the mutual information $I(Z_1; Z_2)$ with those of the structural regularization methods. Typically, we set $\lambda = 0.1$ for comparison with discretized and sparse projectors. As shown in Table 5, we find that even if $I(Z_1; Z_2)$ of the bottleneck regularizer is lower than those of structural regularization methods, it has an inferior lower bound and poorer downstream performance due to low $I(Z_1; R)$, which accords with our conjecture.

Table 5: Comparison between Training and Structural Regularization.

| Method | $I(Z_1; Z_2)$ | $I(Z_1; R)$ | lower bound | ACC |
|---|---|---|---|---|
| Bottleneck Regularizer ($\lambda = 10^{-4}$) | 1.28 | -3.70 | -4.98 | 59.38 |
| Bottleneck Regularizer ($\lambda = 0.1$) | 0.36 | -5.39 | -5.75 | 50.72 |
| Discretized Projector | 1.16 | -3.52 | -4.68 | 60.16 |
| Sparse Projector | 1.02 | -3.49 | -4.51 | 61.99 |

### B.10 GENERALIZATION OF OUR METHODS TO SUPERVISED LEARNING

In this section, we generalize our training and structural regularization methods to supervised learning. To elaborate, we add a new single-layer classifier to the end of the projection head, where it has access to the ground truth labels of images. During training, we simply shift the initial InfoNCE loss to the cross entropy loss of the new classifier. Note that the other classifier attached to the end of the encoder remains and our accuracy is measured from this classifier instead of the new one. We collect the classification accuracy of SimCLR on CIFAR-100. In Table 6, we find that both adding a regularization term (with the coefficient $\lambda = 10^{-4}$) and using a discretized projector (with discrete point number $L = 10$) outperforms the baseline, indicating that our proposed methods can be well generalized to the supervised learning setting.

Table 6: Classification accuracy of the features before the projector in supervised learning.

| Method | Baseline | Bottleneck Regularizer | Discretized Projector |
|---|---|---|---|
| ACC | 71.85 | 72.65 | 72.85 |

### B.11 SUPPLEMENTARY RESULTS WITH NONLINEAR CLASSIFIERS

In order to further verify that the downstream performance is bottlenecked by $I(Z_1; Z_2)$ instead of the capability of the classifier, we conduct additional experiments and test the downstream accuracy of our methods with nonlinear classifiers. To be more specific, we finetune an offline classifier for 200 epochs after pretraining under the SimCLR framework on CIFAR-100. We set the coefficient $\lambda = 10^{-4}$ for the bottleneck regularizer, discrete point number $L = 30$ for the discretized projector, and activated features $k = 5 \times 10^3$ for the sparse projector. As displayed in Table 7, the accuracies with both linear and nonlinear classifiers improve with the use of our methods, which indicates that it is not the classifier but the mutual information $I(Z_1; Z_2)$ that restrains the downstream performance.

Table 7: Fine-tuning accuracy of ResNet-18 pretrained by SimCLR with the original and our proposed projectors on CIFAR-100.

| Method | Baseline | Bottleneck Regularizer | Discretized Projector | Sparse Projector |
|---|---|---|---|---|
| Finetuning ACC | 72.31 | 72.88 | 73.27 | 73.48 |

### B.12 COMBINATIONS OF DIFFERENT REGULARIZATION METHODS

Given that we have achieved a certain amount of improvement by means of training and structural regularization, it is intriguing to investigate whether the combination of several of the proposed methods can yield even better performance. With the default parameters, we conduct experiments using different combinations and collect the downstream accuracy of SimCLR on CIFAR-100, which are listed in Table 8. We observe that combining any two of our proposed methods can bring further improvement to downstream performance.

Table 8: Linear probing performance of the combination of our proposed projectors on CIFAR-100. BR: bottleneck regularizer; DP: discretized projector; SP: sparse projector.

| Method | BR | DP | SP | BR+DP | BR+SP | DP+SP | BR+DP+SP |
|---|---|---|---|---|---|---|---|
| ACC | 59.38 | 60.16 | 61.99 | 60.68 | **63.23** | 62.42 | 62.49 |

## C THEORETICAL VALIDATION ON THE DISCRETIZATION METHOD

In this section, we provide a proof for theorem 4.1 which gives a theoretical illustration for the discretized projector.

*Proof.* We consider the trade-off between $I(Z_1; R)$ and $I(Z_1; Z_2)$.

$$
\begin{aligned}
I(Z_1; R) - I(Z_1; Z_2) &= H(R) - H(R|Z_1) - H(Z_2) + H(Z_2|Z_1) \\
&= H(R) - H(Z_2) - (H(R|Z_1) - H(Z_2|Z_1)) \\
&= H(R) - H(Z_2) - (H(R|Z_2) - H(Z_2|R, Z_1)) \\
&= -H(Z_2) + I(Z_2; R) + H(Z_2|R, Z_1) \\
&\geq -H(Z_2) + I(Z_2; R)
\end{aligned}
$$

It remains to be shown that $H(R|Z_1) - H(Z_2|Z_1) = H(R|Z_2) - H(Z_2|R, Z_1)$ holds. According to lemma A.1, we know that $Z_1$ and $R$ are independent random variables when given $Z_2$, i.e., $H(R|Z_2) = H(R|Z_2, Z_1)$. Therefore, we have

$$
\begin{aligned}
H(R|Z_2) - H(Z_2|R, Z_1) &= H(R|Z_2, Z_1) - H(Z_2|R, Z_1) \\
&= (H(R, Z_2|Z_1) - H(Z_2|Z_1)) - H(Z_2|R, Z_1) \\
&= (H(R, Z_2|Z_1) - H(Z_2|R, Z_1)) - H(Z_2|Z_1) \\
&= H(R|Z_1) - H(Z_2|Z_1).
\end{aligned}
$$

Thus the proof of theorem 4.1 is finished and it provides a theoretical perspective to understand how the discretized projector works and why there is a peak in downstream performance when the discrete point number grows from one to infinity. □

