# OpenReview forum: "Projection Head is Secretly an Information Bottleneck"
_ICLR.cc/2025/Conference — ICLR 2025 Poster_

### Official Review · Reviewer_7oxV · 2024-10-25

**Soundness:** 2
**Presentation:** 3
**Contribution:** 2
**Rating:** 5
**Confidence:** 4

**Summary:**

This paper aims to understand the role of the projection head in contrastive learning from an information-theoretic perspective. Treating the mutual information between the supervised label Y (potentially in downstream tasks) and the learned representations Z1 (before the projection head) as the metric for downstream task accuracy, this paper presents both the lower and upper bounds based on mutual information between features at different layers and proposes that the term I(Z1;Z2) as the information gap plays an important role. To boost the performance of downstream tasks, this paper proposes two types of modified loss functions to regularize the term I(Z1;Z2).

**Strengths:**

1. It is novel to view the projection head as information bottle neck, which provides one way to regularize the infoNCE loss to directly boost the downstream performance
2. Different types of regularization are proposed in this paper, which enables the cross-comparison and validates the consistent improvement over the canonical infoNCE loss.

**Weaknesses:**

1. The performance of downstream task (e.g. classification accuracy) is related to mutual information implicitly. Thus, using mutual information between Y and Z1 as the target to justify the claim seems insufficient to me. It would be very helpful to provide concrete toy examples to relate the downstream accuracy and the mutual information in an explicit way. In this sense, the information gap I(Z1;Z2) seems to be a reasonable motivation for the regularized loss, but is not strong enough as a theoretical explanation to fully understand the role of the projection head.
2. The message conveyed in experiments is not clear enough.
	- For example, in Figure 2 and 3, instead of the comparison between lower bound VS downstream accuracy, as the claim of this paper is the role of I(Z1;Z2), it would be more insightful to plot downstream accuracy VS I(Z1;Z2).
	- In addition, in Figure 4, it would be more helpful to compare the proposed methods in terms of the downstream accuracy with more datasets to see if the ranks of all methods remain consistent.

**Questions:**

1. [Theorem 3.1 and 3.2] Regarding the lower and upper bounds for the mutual information I(Y,Z1) as the proxy of the downstream accuracy, there are three terms in each bound, in which, the effect of I(Z1;R) (or I(Y;Z2)) and I(Z1;Z2) are mixed. Also, as  is shown in Figure 2, both the upper and lower bounds are not tight, it is not sufficiently convincing to claim that the role of the projection head is only revealed via I(Z1;Z2). I was wondering if we could design experiments (e.g. fixing the encoder) such that other terms in the bounds are fixed to study the role of I(Z1;Z2).
2. [Calculations of mutual information] Suppose there exist perfect features Z1 such that Z1 = Y almost surely. This corner case with infinity mutual information should be taken care of.
3. [Figure 3] It seems that points in the right-top corner are more close to a horizontal line. Is it because the upper and lower bounds are not tight enough?
4. In addition, there are several related papers investigating the role of the projection head:
	- Saunshi, Nikunj, Jordan Ash, Surbhi Goel, Dipendra Misra, Cyril Zhang, Sanjeev Arora, Sham Kakade, and Akshay Krishnamurthy. "Understanding contrastive learning requires incorporating inductive biases." In International Conference on Machine Learning, pp. 19250-19286. PMLR, 2022.
	- Gui, Yu, Cong Ma, and Yiqiao Zhong. "Unraveling Projection Heads in Contrastive Learning: Insights from Expansion and Shrinkage." arXiv preprint arXiv:2306.03335 (2023).
	- Wen, Zixin, and Yuanzhi Li. "The mechanism of prediction head in non-contrastive self-supervised learning." Advances in Neural Information Processing Systems 35 (2022): 24794-24809.
	- Xue, Yihao, Eric Gan, Jiayi Ni, Siddharth Joshi, and Baharan Mirzasoleiman. "Investigating the Benefits of Projection Head for Representation Learning." arXiv preprint arXiv:2403.11391 (2024).

I am open to raising my score if the more explicit connection between the mutual information I(Y;Z1) and the downstream accuracy can be presented in some concrete settings, and if fine-grained analysis can be provided to study the tightness of the lower and upper bounds.

---

> ### Author Response · Authors · 2024-11-22
> **Response to Reviewer 7oxV (1/2)**
>
> We sincerely thank Reviewer 7oxV for a critical review of our paper and have carefully addressed your concerns. We will elaborate on the more explicit connection between $I(Y;Z_1)$ and the downstream accuracy and provide supplemental fine-grained analysis on the tightness of the lower and upper bounds.
>
> ---
>
> Q1. Clarification on the connection between $I(Y;Z_1)$ and the downstream accuracy.
>
> A1. In fact, although the connection between $I(Y; Z_1)$ and the downstream accuracy is implicit, **the connection between $I(Y ;Z_1)$ and the downstream classification loss (Cross-Entropy Loss) is explicit**. To be specific, according to the definition of mutual information, we know that $I(Y; Z_1)=H(Y)-H(Y|Z_1)$ and $H(Y)$ is a constant. So we focus on the connection between $H(Y|Z_1)$ and the Cross-Entropy (CE) loss. During the downstream evaluation process, the CE loss is
>
> $L_{CE}(\theta,Z_1,Y) = -\sum\limits_{Z_1,Y}p(Y|Z_1)\log q(Y|Z_1,\theta)$,
>
> where $\theta$ are the parameters of the downstream classifier following the encoder features $Z_1$, $p(Y|Z_1)$ is the ground-truth distribution of labels and $q(y|Z_1,\theta)$ is the distribution estimated by the downstream classifier. In the information theory [1], we have the following equation
>
> $L_{CE}(\theta,Z_1,Y) = H(Y|Z_1) + D_{\rm KL}(p(Y|Z_1)q(Y|Z_1,\theta))$,
>
> where $D_{KL}$ is the KL-divergence. The equation implies that when $\theta$ is optimal (denoted as $\theta^\star$), the downstream classification loss is equal to $H(Y|Z_1)$. **Consequently, when we obtain a larger $I(Z_1;Y)$, we obtain a smaller $H(Y|Z_1)$, which means a small downstream classification cross-entropy loss $L_{CE}(\theta^\star, Z_1, Y)$.** And the connection between the classification loss and the classification accuracy is explicit.  Consequently, there exists a strong explicit connection between $I(Y;Z_1) and the downstream accuracy.
>
> Reference
>
> [1] Thomas, M. T. C. A. J., and A. Thomas Joy. Elements of information theory. Wiley-Interscience, 2006.
>
> ---
>
> Q2. Investigating the role of $I(Z_1;Z_2)$ with fixed $I(Z_1;R)$.
>
> A2. Thanks for your suggestions! In supplemental experiments, we sort the data points by the values of $I(Z_1,R)$ and find data points that have similar $I(Z_1, R)$ values (the gap is smaller than 0.01). We then observe the correlation between $I(Z_1, R)$ and the linear accuracy. As shown in Figure 9 of the revision, with similar $I(Z_1; R)$ values, the downstream performance increases as I(Z1; Z2) decreases (the average Pearson correlation is -0.93), which further verifies the effectiveness of our theoretical bounds and the regularization on mutual information.
>
> ---
>
> Q3. It would be more helpful to compare the proposed methods in terms of the downstream accuracy with more datasets to see if the ranks of all methods remain consistent.
>
> A3. Thanks for the suggestion. We have added supplemented experiments on CIFAR-10 and ImageNet-100 in Appendix As shown in Figure 8, the rank of the three methods shows differences on different datasets (e.g., Sparse Projector performs best on ImageNet-100 while Discretized Projector performs best on CIFAR-10). However, the rank of accuracy is consistent with the rank of the lower bounds, which further verifies the tightness of our theoretical estimation.
>
> ---
>
> Q4. Suppose there exist perfect features Z1 such that Z1 = Y almost surely. This corner case with infinity mutual information should be taken care of.
>
> A4. Thanks for pointing this out. To address this corner case, we follow [1], [2] and supplement a common assumption in the theoretical analysis of contrastive learning, i.e., the set of samples is a finite but exponentially large set, which means $X$ is discrete. Then we can obtain that $Z_1$ is discrete and the mutual information will not be infinite. We have added this setting in the revision (Section 3.1)
>
> **Reference:**
>
> [1] Haochen, et al. "Provable Guarantees for Self-Supervised Deep Learning with Spectral Contrastive Loss." NeurIPS 2021.
>
> [2] Wang, Yifei, et al. "Chaos is a ladder: A new theoretical understanding of contrastive learning via augmentation overlap." ICLR 2022.
>
> ---
>
> Q5.  It seems that points in the right-top corner are more close to a horizontal line. Is it because the upper and lower bounds are not tight enough?
>
> A5. We believe that it results from that we do not sample sufficient data points. In the revision, We have supplemented additional experiments by training different training parameters and updated Figure 3. To be specific, we respectively use different learning rates, different weight decays, different temperature parameters, and different augmentation strengths of ColorJitter. As shown in Figure 3, the overall correlation is quite strong (the average Pearson correlation is 0.7875), which further verifies that our theoretical estimation of the downstream performance of the encoder features is tight.
>
> ---

---

> ### Author Response · Authors · 2024-11-22
> **Response to Reviewer 7oxV (2/2)**
>
> Q6. Additional related works.
>
> A6. Thanks for mentioning those works. We have added them to our paper.
>
> ---
>
> Thank you again for your critical reading and insightful suggestions. If you find our response satisfactory, we would appreciate it if you could re-evaluate our work based on these updated results. We are happy to address any remaining concerns.

---

### Official Review · Reviewer_7XZ7 · 2024-10-26

**Soundness:** 3
**Presentation:** 3
**Contribution:** 2
**Rating:** 8
**Confidence:** 3

**Summary:**

The paper analyzes the performance of encoder features (before projection head) in contrastive learning. Using an information-theoretic perspective, it derives bounds on the mutual information of the encoder features with the labels. Based on these bounds, it concludes that a good projection head removes unnecessary information from encoder features. It then proposes training and structural modifications to this end, and validate the effectiveness empirically on a few datasets.

**Strengths:**

The paper provides a novel analysis of pre-projection features in contrastive learning based on mutual information. The proposed modifications to contrastive learning are well motivated by the theory, and the experiments help validate the theory.

The paper is overall pretty easy to follow.

**Weaknesses:**

See questions.

**Questions:**

I wonder if/how the analysis can be applied to supervised learning, e.g. could features from before the last layer in supervised learning also be improved in a similar fashion?

What is the impact/motivation for the choice of $\alpha$ in the matrix based mutual information?

---

> ### Author Response · Authors · 2024-11-22
> **Response to Reviewer 7XZ7**
>
> We thank Reviewer 7XZ7 for appreciating our paper. Below we will address your questions point by point.
>
> ---
>
> Q1. Could features from before the last layer in supervised learning also be improved in a similar fashion?
>
> A1. An interesting question! We supplement additional experiments on supervised learning. To be specific, we respectively add the projectors used in SimCLR (the original projector and our proposed changes) after the ResNet-18 backbone and then train the networks with supervised learning. During the pretraininig process, we use the default parameters of the projectors used in SimCLR. For the evaluation, we discard the projector and evaluate the classification accuracy of the backbones.
>
> *Table 1. Classification accuracy of the features before the projector in supervised learning.*
>
> | Method | Accuracy |
> | --- | --- |
> | baseline | 71.9 |
> | bottleneck regularization | 72.7 (+0.8) |
> | discretized projector | **72.9 (+1.0)** |
>
> As shown in the table above, the proposed changes have exhibited improvements with the default parameters of projectors in SimCLR, which further verifies the effectiveness and potential of our findings.
>
> ---
>
> Q2. What is the impact/motivation for the choice of α in the matrix based mutual information?
>
> A2. We mainly follow the implementation in [1], where the authors observe that using $\alpha=2$ and $\alpha=1$ almost does not influence the estimation accuracy but training with $\alpha=1$ is much more time-consuming (because of Singular Value Decomposition). Besides, we also conduct experiments to observe the influence of choice of $\alpha$ on the performance of the training regularizers. As shown in the following table, we find that taking $\alpha=2$ achieves the best empirical performance.
>
> *Table 1. Linear probing accuracy of the models trained with bottleneck regularization (implemented by selecting different $\alpha$) on CIFAR-100.*
>
> | alpha | 1 | 2 | 3 |
> | --- | --- | --- | --- |
> | acc | 59.06 | 59.38 | 58.59 |
>
> **Reference:**
>
> [1] Tan, et al. "Information Flow in Self-Supervised Learning." ICML 2024.
>
> ---
>
> Thank you again for your rating and questions. Please let us know if there is more to clarify.

---

### Official Review · Reviewer_7rGv · 2024-10-31

**Soundness:** 3
**Presentation:** 3
**Contribution:** 3
**Rating:** 8
**Confidence:** 4

**Summary:**

- The paper proposes that contrastive methods using separate representation layers (for downstream tasks) and projection heads (for computing the contrastive loss) are actually taking advantage of an Information Bottleneck between the representation layer and the projection head.
- The authors derive lower and upper bounds on I(Y;Z_1) (the Mutual Information between the downstream target, Y, and the representation Z_1) that show an I(Z_1;Z_2) term being minimized (where Z_2 is the projection layer’s representation).
- The authors show a correlation between their bounds and the validation set accuracy to attempt to justify their theoretical claims.
- The authors then use an estimator of I(Z_1;Z_2) as a regularizer in the contrastive loss and show empirically that using that regularizer improves performance on the downstream classification task for three image classification problems.
- The authors then compare with two previously-studied “structural” regularizers that (they argue) also minimize I(Z_1;Z_2) and show that those regularizers also improve performance on these classification tasks.

**Strengths:**

- The writing is generally clear and easy to follow.
- The paper gives an Information Theoretic explanation for why the projection head is important in many contrastive loss settings.
- The paper proposes a method to directly minimize the claimed quantity of interest, I(Z_1;Z_2) and shows that using that minimizer can improve accuracy on the downstream classification task.
- The paper provides an explanation for the effectiveness of aggressive bottlenecking between the representation layer and the projection head.

**Weaknesses:**

- Line 154: The simplest relationship of interest is that I(Y;Z_1) >= I(Y;Z_2), by the data processing inequality, so maximizing I(Y;Z_2) also maximizes I(Y;Z_1), which you acknowledge as the quantity of interest. Since by assumption we don’t have access to Y, we are limited to maximizing I(X;Z_2), which is what many SSL objectives do (including InfoNCE); again by the data processing inequality, I(X;Z_1) >= I(X;Z_2). But in the limit when I(X;Z_2) approaches H(X), I(Y;Z_2) must be maximized. While my view is that correct compression is important for learning good representations, this kind of information maximization argument is common in SSL (since you don't know what the downstream task is, why would you want to throw out any information in the representation?), so I think it is important to be careful how you justify your approach.
- Thm 3.2, line 209: You appear to be using I(Y;Z_1|Z_2) = H(Z_1|Z_2) - H(Z_1|Y,Z_2) and then assuming that H(Z_1|Y,Z_2) >= 0, but that is not guaranteed unless Z_1 is discrete, which it probably isn’t. So it seems to me that the bound is broken, or at least that your proof of the bound may be incorrect. Have I misunderstood your justification for the inequality in line 209?
- Figure 2a: The lower bound estimate is non-monotonic with a noticeable (but small) peak at 80 epochs, even though validation accuracy increases monotonically through epoch 100. This indicates that you should probably run more experiments to get more robust statistical results, or that your analysis is not perfectly accurate. Or perhaps some other explanation, but it doesn’t appear to be addressed in the paper, as you claim (line 264) that the lower bound “continuously increase[s]”.
- More generally, the “Tendency during training” paragraph is weak, as there are many quantities that can generally increase during training while accuracy is increasing, without real explanatory power, such as weight magnitudes.
- The two “structural” regularizers were already known to improve downstream task performance, so it is not clear to me that the empirical results are adding much to the presentation, although they do show that (in the settings you consider) the explicit minimization of I(Z_1;Z_2) is less effective than just using a strong architectural bottleneck.
- It might be helpful in Figure 1 to show how Z_1 is used in downstream tasks – i.e., add an edge Z_1 \to \hat{Y} or something like that.
- Line 152: “pipline” => “pipeline”.

**Questions:**

- Line 250: why do you set \alpha=2? Wouldn’t \alpha=1 correspond to the Shannon Mutual Information? Is there some benefit to distorting the Shannon Information in this setting?
- Given that the “structural” regularizers are more effective than the direct minimization of (an estimate of) I(Z_1;Z_2), my sense is that this paper needs to do more work to demonstrate that the “structural” regularizers improve performance for the stated reason (that they minimize I(Z_1;Z_2)). I don’t contest that they *are* reducing I(Z_1;Z_2), but I would like to know that there isn’t some alternative hypothesis that better explains the improvement in performance. Alternatively, perhaps it’s worth considering a variational approach to minimizing I(Z_1;Z_2), such that you can properly bound I(Z_1;Z_2) – perhaps the relatively poor performance of your objective-based approach is due to using an estimate of I(Z_1;Z_2) rather than a proper bound? All that being said, I think the underlying hypothesis is plausible, so I welcome your pushback on my concerns about the paper. I’m open to good arguments to increase my rating.

---

> ### Author Response · Authors · 2024-11-22
> **Response to Reviewer 7rGv (1/2)**
>
> We thank Reviewer 7rGv for careful reading and critical review.  Below we will address your main concerns in detail.
>
> ---
>
> Q1. When I(X;Z_2) approaches H(X), I(Y;Z_2) must be maximized, which maximizes I(Y;Z_1). This kind of information maximization argument is common in SSL. so I think it is important to be careful how you justify your approach.
>
> A1. Indeed, the perspective that optimizing the contrastive objective can maximize $I(Y;Z 2)$ has been well studied. **However, we should note that only the lower bound $(I(Y;Z_1) >= I(Y;Z_2))$ can not guarantee that maximizing $I(Y; Z2)$ also maximizes $I(Y;Z_1)$.** For example, we follow the default settings in Figure 3 and observe the correlation between the linear accuracy of encoder features ($Z_1$) and projector features ($Z_2$). As shown in Figure 10 in the revision, the correlation is quite weak (Pearson Correlation is smaller than 0.5), which means that $I(Y;Z_2)$ is not an accurate estimation of $I(Y;Z_1)$ and we can not obtain that “maximizing I(Y;Z_2) also maximizes I(Y;Z_1)”.
>
> Instead, we should consider a tighter lower bound and an upper bound on $I(Y;Z_1)$. Theorems 3.1 and 3.2 show that it is decided by how we design a projector. In Figure 3, we show that the correlation between our bounds and the downstream accuracy is significantly stronger than 0.5, which implies that our bounds are tighter and can provide more meaningful insights on how to improve the performance of encoder features than simply ignoring the influence of different projectors.
>
> ---
>
> Q2. You appear to be using I(Y;Z_1|Z_2) = H(Z_1|Z_2) - H(Z_1|Y,Z_2) and then assuming that H(Z_1|Y,Z_2) >= 0, but that is not guaranteed unless Z_1 is discrete, which it probably isn’t.
>
> A2.  Thanks for pointing this out! In fact, we need an assumption that the set of samples is a finite but exponentially large set, i.e., $X$ is discrete. Then we can obtain that $Z_1$ is discrete and the bound is not broken.  **We note that it is a common assumption in the theoretical analysis of contrastive learning ([1], [2]).** We have added this setting in the revision (Section 3.1, Line 145).
>
> **Reference:**
>
> [1] Haochen, et al. "Provable Guarantees for Self-Supervised Deep Learning with Spectral Contrastive Loss." NeurIPS 2021.
>
> [2] Wang, Yifei, et al. "Chaos is a ladder: A new theoretical understanding of contrastive learning via augmentation overlap." ICLR 2022.
>
> ---
>
> Q3. The lower bound estimate is non-monotonic with a noticeable (but small) peak at 80 epochs, even though validation accuracy increases monotonically through epoch 100. This indicates that you should probably run more experiments to get more robust statistical results, or that your analysis is not perfectly accurate.
>
> A3.Thanks for your suggestions. To get more robust statistical results, **we have supplemented additional experiments of SimCLR and Barlow Twins on CIFAR-10, CIFAR-100, and ImageNet-100.** As shown in Figure 5, and Figure 6, we observe that the tendency of both bounds remains consistent with classification accuracy in various datasets and different methods. Combined with our other results (the evaluation of correlation in Figure 3 and Figure 4), we believe that there is a strong correlation between our theoretical guarantees and downstream performance.
>
> ---
>
> Q4.  More generally, the “Tendency during training” paragraph is weak, as there are many quantities that can generally increase during training while accuracy is increasing, without real explanatory power, such as weight magnitudes.
>
> A4. Indeed, the experiment “tendency during training” is just a part of our empirical results to verify the relationship between downstream accuracy and the theoretical bounds.  To draw the final conclusion, it is necessary to analyze the results combined with Figure 3 and Figure 4. The consistent results can verify that our theoretical bounds are accurate estimates of downstream performance.

---

> ### Author Response · Authors · 2024-11-22
> **Response to Reviewer 7rGv (2/2)**
>
> Q5. The two “structural” regularizers were already known to improve downstream task performance, so it is not clear to me that the empirical results are adding much to the presentation.
>
> A5. In fact, to the best of our knowledge, these two designs are not designed to improve downstream classification performance. To be specific, the papers related to Sparse Autoencoders mainly focus on enhancing the interpretability of features and usually hurt the downstream performance ([1], [2]). And the discretization techniques are designed to save the memory and accelerate the inference ([3]). Both of them do not show improvements in downstream classification performance. So we believe that adopting them to increase classification performance is novel. The effectiveness of bottleneck architectures contributes to verifying our claims that an effective projector should be an information bottleneck.
>
> Reference:
>
> [1] Gao, Leo, et al. "Scaling and evaluating sparse autoencoders." arXiv preprint arXiv:2406.04093 (2024).
>
> [2] Cunningham, Hoagy, et al. "Sparse autoencoders find highly interpretable features in language models." arXiv preprint arXiv:2309.08600 (2023).
>
> [3] Gholami, Amir, et al. "A survey of quantization methods for efficient neural network inference." Low-Power Computer Vision. Chapman and Hall/CRC, 2022. 291-326.
>
> ---
>
> Q6.  It might be helpful in Figure 1 to show how Z_1 is used in downstream tasks – i.e., add an edge Z_1 \to \hat{Y} or something like that.
>
> A6.  Thanks for your suggestion and we have added the edge between $Z_1$ and $\hat{Y}$ in the revision.
>
> ---
>
> Q7. The typo in Line 152.
>
> A7. Thanks for pointing it out. We have fixed it in our revision.
>
> ---
>
> Q8. Why do you set \alpha=2? Wouldn’t \alpha=1 correspond to the Shannon Mutual Information?
>
> A8. We mainly follow the implementation in [1], where the authors observe that using $\alpha=2$ and $\alpha=1$ almost does not influence the estimation accuracy but training with $\alpha=1$ is much more time-consuming (because of Singular Value Decomposition). Besides, we also conduct experiments to observe the influence of choice of $\alpha$ on the performance of the training regularizers. As shown in the following table, we find that taking $\alpha=2$ achieves the best empirical performance.
>
> *Table 1. Linear probing accuracy of the models trained with bottleneck regularization (implemented by selecting different $\alpha$) on CIFAR-100.*
>
> | alpha | 1 | 2 | 3 |
> | --- | --- | --- | --- |
> | acc | 59.06 | 59.38 | 58.59 |
>
> **Reference:**
>
> [1] Tan, et al. "Information Flow in Self-Supervised Learning." ICML 2024.
>
> ---
>
> Q9. Why structural regularization performs better than training regularization.
>
> A9. As shown in Theorem 3.1, an effective projector should reduce $I(Z_1; Z_2)$ while maximizing $I(Z_1;R)$. which indicates that when we apply regularizations on $I(Z_1;Z_2)$, we should try to avoid decreasing $I(Z_1, R)$. As shown in the following table, by comparing training and structural regularization, we find that structural regularizations can preserve $I(Z_1; R)$ better.
>
> *Table 2. The comparison of $I(Z_1; R) between the training regularizer and structural regularizers on CIFAR-100.*
>
> | Method | $I(Z_1;Z_2)$ | $I(Z_1;R)$ | lower bound | ACC |
> | --- | --- | --- | --- | --- |
> | Training Regularizer ($\lambda=0.0001$) | 1.28 | -3.70 | -4.98 | 59.38 |
> | Discretized Projector | 1.16 | -3.52 | -4.68 | 60.16 |
> | Sparse Projector | 1.02 | -3.49 | -4.51 | 61.99 |
> | Training Regularizer ($\lambda=0.1$) | 0.36 | -5.39 | -5.75 | 50.72 |
>
> **We note that even when we adopt mild training regularizations, i.e., the mutual information $I(Z_1,Z_2)$ after the training regularization is larger than structural regularization, $I(Z_1,R)$ of the training regularization is still much smaller,** which indicates a smaller lower bound and inferior performance.  Moreover, when we enhance the strength of regularization, $I(Z_1;R)$ drops sharply and leads to much inferior performance. A reasonable explanation for these results is that the specially designed architectures, e.g., the topK function in Sparse autoencoders, can preserve the most important information and discard unnecessary information while the simple training regularization takes a disadvantage on that.
>
> ---
>
> Thank you again for your thoughtful insights and advice. We respectfully suggest that you could re-evaluate our work based on these updated results. We are very happy to address any remaining concerns.

---

> > ### Comment · Reviewer_7rGv · 2024-11-22
> >
> > Thanks for the thorough response to my review! With your explanations and new results, I am happy to increase my recommendation.

---

### Official Review · Reviewer_DcxU · 2024-11-01

**Soundness:** 3
**Presentation:** 3
**Contribution:** 3
**Rating:** 6
**Confidence:** 3

**Summary:**

This paper studies the design choices of the projection head in contrastive learning, and shows that the projection head should serve as a proper information bottleneck that keeps only "useful" information related to the class label or the contrastive objective.

For the problem setup, consider the information flow of $Y \rightarrow X \rightarrow Z_1 \rightarrow Z_2 \rightarrow R$, where $Y$ is the semantic info (e.g. class label), $X$ is the observable/input, $Z_1$ is the pre-projection head features, $Z_2$ is the projection head output, and $R$ is the contrastive label.
The paper argues that we should focus on $I(Y; Z_1)$ (rather than $I(Y; Z_2)$), since the projection head is discarded in downstream training.

The main results are as follows:
- It provides **upper and lower bound on the downstream performance** for the pre-projection head features.
  - Lower bound: $I(Y;Z_1) \geq I(Z_1; R) - I(Z_1; Z_2) + \text{const}$, where $\text{const} = I(R;Y)$ is not affected by the learned features.
  - Upper bound: $I(Y; Z_1) \leq I(Y; Z_2) - I(Z_1; Z_2) + H(Z_1)$.
- We want to adjust the terms on the right hand side so that the bounds are tighter, which leads to **two modifications**: 1) adding a mutual-information-based regularization term in the training objective, and 2) modifying the projector architecture.
  - The **mutual information regularizer** $I(Z_1; Z_2)$: estimated using the matrix mutual information (Renyi entropy) as the surrogate. The matrices are $\hat{Z}_1\hat{Z}_1^\top, \hat{Z}_2\hat{Z}_2^\top$, where $\hat{Z}_1, \hat{Z}_2$ denote the normalized feature matrix before and after the projection head.
    - Result: The accuracy improvements are slightly higher for Barlow Twins than for SimCLR.
  - 2 types of **structural regularization**, both meant to bottleneck the information allowed by the projection head.
    - Discretized projector: the output is discretized into $L$ integer values.
    - Sparse projector: using the top-k activation function, which keeps the $k$ largest hidden representation.
        - The experiments use $k=0.001d$ or $k=0.2d$, where $d$ is the projector's latent dimension.

**Strengths:**

- The proposed changes are motivated by the terms in the mutual information-based lower and upper bounds.
- The proposed changes lead to an 0.26 to 3.99 increase in the accuracy of SimCLR and Barlow Twins on CIFAR-10, CIFAR-10, and ImageNet-100.

**Weaknesses:**

Since the theoretical results of the paper are not strong enough to serve as a major contribution, the empirical results should be strong.

However, the relation between the derived bounds and the empirical results are not strong: the correlation is computed based on a small number of points (please see the question below), and there is no correlational result on Barlow Twins.

**Questions:**

- Fig 2: the curves are only shown up to 100 epochs, at which point the validation accuracy hasn't saturated. What happens to epoch 100-200 (Sec 4 mentions that the models are trained for 200 epochs on CIFAR-100)?
- Do you observe the same trends on other datasets (i.e. CIFAR-10 and ImageNet-100, which are partially shown in Fig 3) and from Barlow Twins?
- Fig 3: are these results varying across training hyperparameters (e.g. learning rate, weight decay, augmentation strengths, temperature of the contrastive objective)?
- For the accuracy: since the paper is based on mutual information, the downstream experiments should also consider nonlinear evaluation, i.e. making the downstream classifier sufficiently powerful to make sure the performance is actually bottlenecked by the "information" in the representation rather than by the classifier.
- Does combining several of the 3 proposed changes lead to further improvements?

---

> ### Author Response · Authors · 2024-11-22
> **Response to Reviewer DcxU**
>
> We thank Reviewer DcxU for a careful review and valuable suggestions. We understand your concerns lie in that we need more experiments to support our analysis. Below we will address your main concerns point by point.
>
> ---
>
> Q1. The tendency of lower and upper bounds and downstream accuracy for 100-200 epochs.
>
> A1. Thanks for pointing this out and we have updated Figure 3 in the revision. As shown in Figure 3, the upper bound continues to rise in 100-200 epochs, which is consistent with the accuracy tendency. For the lower bound, the tendency becomes converged after 100 epochs, which is not perfectly aligned with the accuracy tendency. However, considering the impressive correlation between the lower bound and the downstream accuracy in Figure 3, Figure 4, the tendency of the first 100 epochs, **and especially, additional experiments on the trends on other datasets/methods as your suggestions (Figure 5,6 in the revision), w**e believe that the lower bound is still strongly related to the downstream accuracy and can provide guidance for designing better projectors.
>
> ---
>
> Q2. The trends between lower and upper bounds and downstream accuracy on other datasets and from Barlow Twins.
>
> A2.  Thanks for your suggestions. We have supplemented additional experiments of SimCLR and Barlow Twins on CIFAR-10, CIFAR-100 and ImageNet-100. As shown in Figure 5 and Figure 6 in the revision, we observe that the trends between both bounds and classification accuracy remain fairly high in all datasets and different methods, further indicating that there is a strong relationship between bounds estimation and downstream performance.
>
> ---
>
> Q3. Supplementary results on the correlation between downstream accuracy and theoretical bounds with other training parameters.
>
> A3.  Thanks for your suggestions! We have supplemented additional experiments with different training parameters and updated Figure 3. To be specific, we respectively use different learning rates, different weight decays, different temperature parameters, and different augmentation strengths of ColorJitter. As shown in Figure 3, with additional data points, the correlation is still quite strong (the average Pearson correlation is 0.7875), which further verifies the effectiveness of our theoretical estimation of the downstream performance of the encoder features.
>
> ---
>
> Q4. Supplementary results on nonlinear evaluation.
>
> A4. We understand your concerns and supplement additional experiments on non-linear evaluation. To be specific, taking CIFAR-100 as an example, we follow the default pretraining settings as in Section 4 while fine-tuning the whole backbone and the classifier in downstream classification tasks.
>
> *Table1. Fine-tuning accuracy of ResNet-18 pretrained by SimCLR with the original and our proposed projectors on CIFAR-100.*
>
> |  | finetuning |
> | --- | --- |
> | Baseline | 72.31 |
> | Training regularization | 72.88 (+0.57) |
> | Discretized projector | 73.27 (+0.96) |
> | Sparse projector | **73.48 (+1.17)** |
>
> As shown in the table above, our proposed projectors also outperform the original projector under fine-tuning settings, which further verifies that the performance is actually bottlenecked by the mutual information between encoder and projector features instead of the expressive power of the linear classifier.
>
> ---
>
> Q5. Does combining several of the 3 proposed changes lead to further improvements?
>
> A5.  A good question! We supplement additional experiments by combining three proposed methods. We take CIFAR-100 as an example and follow the default settings in Section 4.
>
> *Table 2. Linear accuracy (%) on CIFAR-100 of the models trained with the projector that adopts the combination of different designs in our paper (BR: bottleneck regularizer; DP: discretized projector; SP: sparse projector.).*
>
> | Baseline | BR | DP | SP | BR + DP | BR+SP | DP + SP | BR + DP +SP |
> | --- | --- | --- | --- | --- | --- | --- | --- |
> | 58.12 | 59.38 | 60.16 | 61.99 | 60.68 | **63.23** | 62.42 | 62.49 |
>
> As shown in the table above, **we observe that combining any two of them can bring further improvements than just one change.** Meanwhile, the aggressive regularization strategy (i.e., the combination of three changes) shows inferior performance. It is reasonable as the projector can not preserve necessary information related to contrastive tasks with too aggressive regularization on the mutual information, which indicates lower downstream performance according to the bounds of performance in Theorem 3.1.
>
> ---
>
> Thank you again for your questions, which help make this work more complete. We have conducted additional experiments and addressed each of your concerns above. We respectfully suggest that you could re-evaluate our work based on these updated results. We are very happy to address your remaining concerns about our work.

---

> ### Author Response · Authors · 2024-11-25
> **Your invaluable input is needed**
>
> Dear Reviewer DcxU,
>
> We have carefully prepared a detailed response to address each of your questions. Would you please take a look and let us know whether you find it satisfactory? Your invaluable input is greatly appreciated. Thank you once again, and we hope you have a wonderful day!
>
> Authors

---

> ### Author Response · Authors · 2024-12-02
> **Your further inputs are greatly appreciated. Only 24 hours left.**
>
> Dear Reviewer DcxU,
>
> For your raised questions, we prepared a detailed response to address your concerns. We were hoping to hear your feedback on them.
>
> As there are only 24 hours left for the reviewer and author discussions and we understand that everyone has a tight schedule, we kindly wanted to send a gentle reminder to ensure that our response sufficiently addressed your concerns or if there are further aspects we need to clarify.
>
> If you could find the time to provide your thoughts on our response, we would greatly appreciate it.
>
> Best, Authors

---

> > ### Comment · Reviewer_DcxU · 2024-12-02
> >
> > Dear authors,
> >
> > I sincerely apologize for my delay!
> >
> > Thank you for the additional experiments and the revision; they have addressed most of my concerns. I've increased my score accordingly.

---

### Meta-Review · Area_Chair_xa32 · 2024-12-23

**Metareview:**

The paper analyzes the performance of encoder features (before projection head) in contrastive learning. Using an information-theoretic perspective, it derives bounds on the mutual information of the encoder features with the labels. Based on these bounds, it concludes that a good projection head removes unnecessary information from encoder features. It then proposes training and structural modifications to this end, and validate the effectiveness empirically on a few datasets.

All reviewers believe this paper made meaningful contributions to contrastive learning. The AC agrees and thus recommends acceptance.

**Additional Comments On Reviewer Discussion:**

There were concerns about the writing and experimental results. The major concerns were addressed in the rebuttal.

---

### Decision · Program_Chairs · 2025-01-22

Accept (Poster)